# Genomic epidemiology of the 2025 mpox epidemic in Sierra Leone

Mpox is a re-emerging zoonotic disease caused by MPXV, which has led to outbreaks across multiple countries in recent years. Sierra Leone reported its first mpox case in 8 years in January 2025, rapidly becoming the epicenter of a continental outbreak with more than 5,000 confirmed cases by August, a surge with unknown origins, timings and drivers. Phylodynamic analyses using 338 genomes generated from 14 districts suggests that the outbreak was caused by lineage G.1 (A.2.2.1) which descended from lineages circulating in Nigeria. Here we observed a strong APOBEC3 mutational enrichment, consistent with sustained human transmission that circulated undetected for ~3 months before the first confirmed case in January 2025. The Western Area Urban district served as the primary hub for nationwide spread and persistence, as well as multiple international export events. We further estimated that the true epidemic size was nearly double official case counts, highlighting substantial surveillance gaps. These findings underscore the urgent need for strengthened genomic and diagnostic surveillance systems across West Africa to pre-empt epidemics.

Mpox is a re-emerging zoonotic disease caused by MPXV, which has driven outbreaks across multiple countries in recent years[1,2]. MPXV is classified into two main clades: I and II[3,4]. Historically, clade I (subdivided into Ia and Ib) was confined to zoonotic infections arising from Central African animal reservoirs, whereas clade II (also subdivided into IIa and IIb) was restricted to West Africa and associated with lower case fatality rates[5].

In the past decade, MPXV has shifted in both epidemiology and evolution, with large outbreaks in both endemic and nonendemic regions, marked by sustained human-to-human transmission, often within sexual networks[6]. Clade IIb/sh2017 lineage A emerged with sustained human transmission in Nigeria in 2014[7,8], giving rise to the global 2022 multi-country outbreak[9,10]. More recently, clade Ib/sh2023 emerged in 2023, causing an outbreak in the eastern Democratic Republic of the Congo (DRC), which was strongly linked to dense sexual transmission networks[11,12]. The viral lineages underlying these epidemics show strong APOBEC3-like mutational biases, a molecular signature characteristic of sustained human-to-human transmission[7,8,11,13].

Sierra Leone (SLE) confirmed its first mpox case in 8 years on 10 January 2025, in a symptomatic man in the capital city, Freetown. The Ministry of Health declared a public health emergency shortly thereafter[14]. From January to July 2025, SLE rapidly became the epicenter of the continental mpox epidemic, accounting for nearly half of Africa's confirmed cases during this period[15]. By 26 September 2025, the country had recorded 5,328 confirmed cases and 56 moxa-associated deaths, with all 16 districts affected[16].

Transmission was concentrated in the Western Area Urban (WAU) and Western Area Rural (WAR) districts. The rapid and explosive outbreak growth in SLE, compared with neighboring countries such as Ghana, Liberia, Guinea and Côte d'Ivoire, likely stems from specific sociodemographic drivers. Freetown's dense and highly mobile population (~1.4 million of the nation's 8.6 million residents) facilitated rapid interdistrict spread, outpacing deployment of interventions such as testing, case isolation and contact tracing, whereas early undetected spread within high-mobility networks likely seeded multiple regions before surveillance was fully activated.

Preliminary genomic analyses indicated that the SLE outbreak derives from clade IIb/sh2017 lineage A.2.2 viruses circulating in Nigeria and neighboring West African countries[17]. Epidemiological data show that the epidemic predominantly affected young adults aged 16–35 years[14,18]. Unlike the early 2022 clade IIb/sh2017 global wave with >90% of cases reported among the men who have sex with men

e-mail: oakcampbell25@gmail.com; johnatsandi@gmail.com; donkumfel@yahoo.co.uk

(MSM) community[9,10,19], and the 2017–2022 Nigerian epidemic in which ~63% of cases were male[20–22], the SLE outbreak exhibits a near-equal sex distribution, few pediatric cases and a skew toward younger adult women and older adult men, patterns consistent with transmission within non-MSM sexual networks[14,18].

This demographic profile more closely resembles clade Ib/sh2023 outbreaks in the DRC, Burundi and Uganda, suggesting that demographic differences across clades may reflect transmission network dynamics rather than viral genetic differences[12,23,24]. However, clades differ in clinical severity: clade Ia has historically been associated with higher case fatality ratios than clade IIb, although recent data suggest that this partly reflects contextual factors, such as malnutrition, co-infections and limited healthcare access, rather than intrinsic viral virulence alone, and the reported mortality rate for clade Ib has remained <1% in most affected countries, comparable to clade IIb[12,25,26]. Recent DRC trial data further highlight that tecovirimat outcomes vary substantially unless patients receive high-quality supportive care[27,28]. The clinical severity of clade IIa in humans remains less well characterized than that of the other subclades.

The limited availability of full-length MPXV genomic data from SLE and neighboring countries has left key questions about transmission dynamics unresolved[29]. The absence of early phase samples, uneven district-level sampling and limited temporal range make it challenging to precisely resolve the origins of the G.1 outbreak, and we cannot exclude the possibility that G.1 emerged elsewhere before introduction to SLE. Here we address these questions using phylogenetic and phylogeographical analysis of 338 MPXV genomes sampled across SLE between 10 January and 3 August 2025, to characterize the origin and spatiotemporal dynamics of the epidemic.

## Results

### New G.1 lineage drives the outbreak in SLE

To investigate the evolutionary relationship between SLE's epidemic lineage and global MPXV diversity, we generated 338 high-quality MPXV genomes from cases across 14 districts between 10 January and 3 August 2025 (Fig. 1a,b), produced through collaboration among the Central Public Health Reference Laboratory (CPHRL, $n = 128$), the Kenema Government Hospital (KGH) Viral Hemorrhagic Fever (VHF) Laboratory ($n = 105$) and the Institut Pasteur de Dakar (IPD) mobile laboratory in Port Loko ($n = 105$). With two additional genomes from the Beijing Institute of Technology[30], the total was 340 high-quality genomes (>60% coverage at 25–8,100× depth), representing 6.9% of all PCR-confirmed cases up to August 2025 (Fig. 1a,b).

We reconstructed the clade IIb phylogeny to determine the relationship between the SLE sequences and the clade IIb/sh2017 lineage. In our phylogeny, 339 of 340 genomes clustered within clade IIb/sh2017, which emerged in humans in southern Nigeria in 2014 and drives the ongoing human epidemic in West Africa[7,8]. According to the nomenclature proposed in refs. 3,4, clade IIb/sh2017 is designated as lineage A (or clade IIb/sh2017/lineage A), with direct descendants designated as, for example, A.1 and subsequent subdivisions as, for example, A.1.1—analogous to the Pango nomenclature used for SARS-CoV-2[31]. Within lineage A, our sequences form a well-supported monophyletic group (100% bootstrap support) descended from lineage A.2.2 (Fig. 1c). In accordance with the nomenclature,

we designated the new SLE lineage as G.1, the alias of A.2.2.1, or clade IIb/sh2017/lineage G.1 (Fig. 1c).

The enrichment—rather than the mere presence—of APOBEC3-context mutations distinguishes sustained human transmission from zoonotic spillover[8]. To test whether G.1 fits this profile, we quantified mutational biases by reconstructing ancestral SNPs across the G.1 phylogeny. Approximately 85% (90 of 106) of reconstructed SNPs were consistent with APOBEC3 editing (TC → TT or GA → AA transitions driven by APOBEC3F[32]), providing strong genomic evidence that G.1 arose through sustained human transmission rather than repeated zoonotic introductions (Fig. 1c and Extended Data Fig. 1).

### Historical circulation of clade IIa in West Africa

We identified one nonclade IIb genome: a clade IIa sequence sampled in mid-January 2025 in WAU. Phylogenetically, it clusters with two clade IIa genomes from Guinea (August and December 2024), forming a sister clade to the 1965 Rotterdam zoo epizootic and near-contemporary museum orangutan sequences[33,34] (Fig. 2). The broader clade encompassing these groups shares a deep common ancestor with the lineage containing the 1958 Copenhagen captive-monkey sequence[35], reinforcing that clade IIa has historically circulated in West Africa. APOBEC3 substitutions were scarce across all three sequences (≤2 per genome), in sharp contrast to the heavily enriched G.1 lineage (Fig. 2 and Extended Data Fig. 1), supporting zoonotic spillover rather than sustained human transmission. Whether this case represents importation from Guinea or an independent zoonotic event in SLE requires additional sampling to resolve.

### G.1 descended from lineages circulating in West Africa

The geographical origin of the G.1 lineage remains unresolved. It is unclear whether it arose directly from endemic clade IIb/sh2017 sublineages in Nigeria or was introduced into SLE via an intermediate source. In our phylogenetic analyses, G.1 is nested within lineage A.2.2, which is primarily sampled from the USA (Fig. 1c). G.1 is separated from its closest relative, a sequence from Togo, by nine APOBEC-like mutations and two non-APOBEC3-like mutations along its stem branch. Previous studies have estimated that APOBEC3-like mutations accumulate at approximately 6 per year (95% confidence interval 5–7), a rate that is consistent across MPXV clades and lineages with sustained human transmission[7,8,12]. The number of APOBEC3-like mutations along the G.1 stem branch suggests that it diverged from its ancestor, the A.2.2 sublineage, approximately 18 months ago. This provides a lower bound on the timing of the lineage's introduction into SLE.

The 11 A.2.2 genomes from the USA were sampled between June 2024 and June 2025 from Illinois (2), California, Massachusetts (2), Georgia, Pennsylvania, Tennessee, Minnesota, Michigan and Virginia, with confirmed travel history to Nigeria for at least 4 of the sequences (Fig. 1c). Based on the long internal branches separating the US A.2.2 genomes, they most likely represent independent viral imports from the Nigerian A.2.2 lineage rather than an established lineage that has been cryptically diversifying in the USA. The closest publicly available Nigerian A.2.2 sequence to the A.2.2 lineage from which the US and G.1 sequences descend is PP853012, sampled in Rivers State in September 2022. Taken together, these findings suggest that lineage A.2.2 in the USA and the G.1 lineage in SLE both descend from the clade IIb/sh2017

**Fig. 1 | Phylogenetic origin and emergence of mpox lineage G.1 in SLE.**
**a**, Confirmed mpox incidence in SLE during 2025, showing the sequencing rate of this study over time (bars) and the time-varying reproductive number (line, right y axis, with 95% CIs shaded in gray). Annotations mark the start of vaccination campaigns in the WAU and WAR, as well as the launch of the national vaccination campaign. **b**, Number of confirmed mpox cases per district in SLE, with color coding by district as shown in the legend. The number of sequences generated per district in this study is annotated in the text. **c**, Clade IIb phylogeny with reconstructed SNPs mapped on to branches and color coded by country

of sampling. We performed ancestral state reconstruction across our clade IIb phylogeny to map SNPs to their corresponding branches. We annotated APOBEC3-characteristic substitutions (CT → TT or GA → AA) in the correct dimer context along branches and calculated their relative proportion across internal branches. APOBEC3 substitutions along the branches are shown in yellow and all other substitutions in gray. Our new sequences are annotated in red and as G.1 in the text. The tree was rooted to the new zoonotic outgroup identified in ref. 7. **d**, Focal G.1 lineage, with sequences color coded by sampling districts in SLE, as shown in the legend.

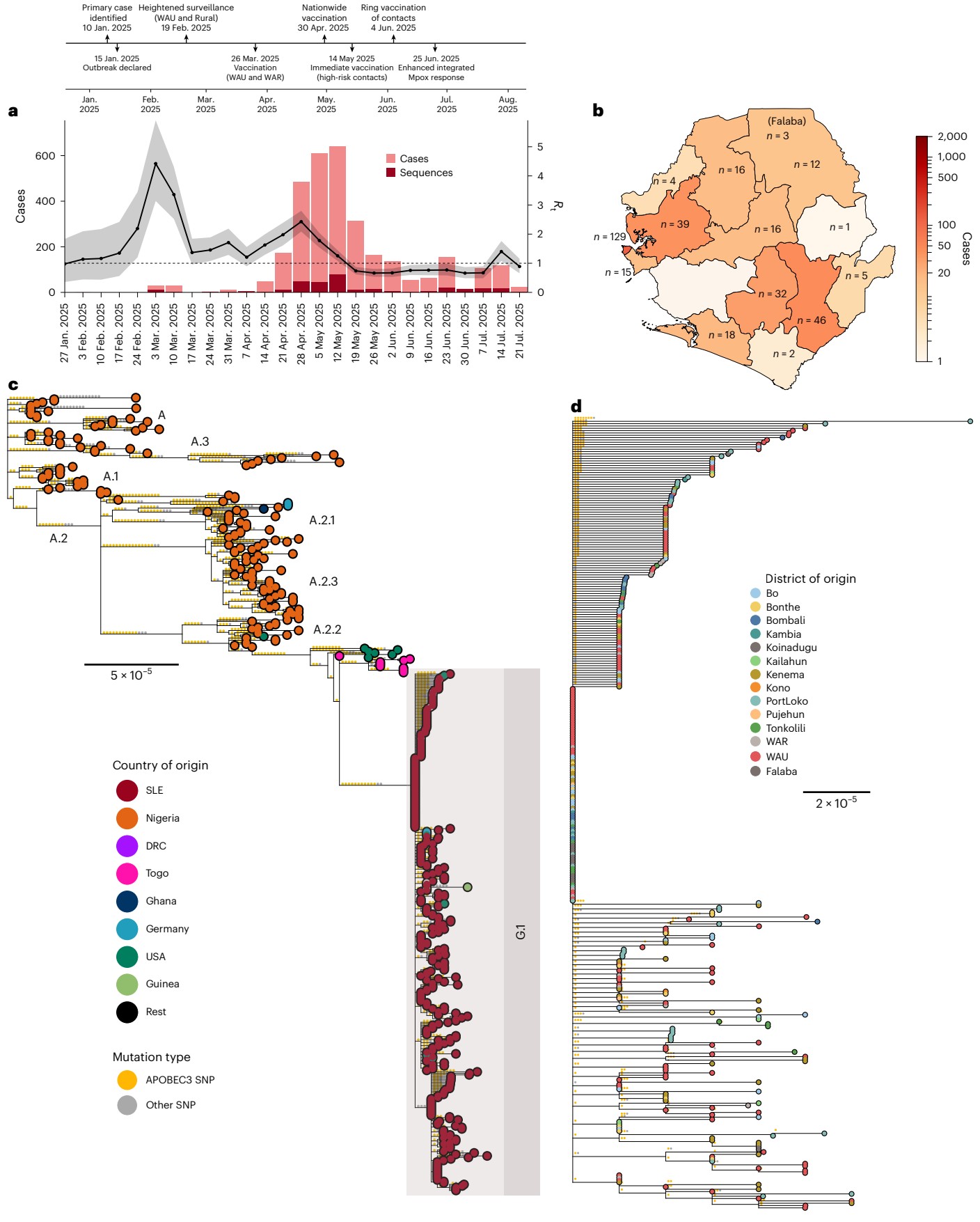

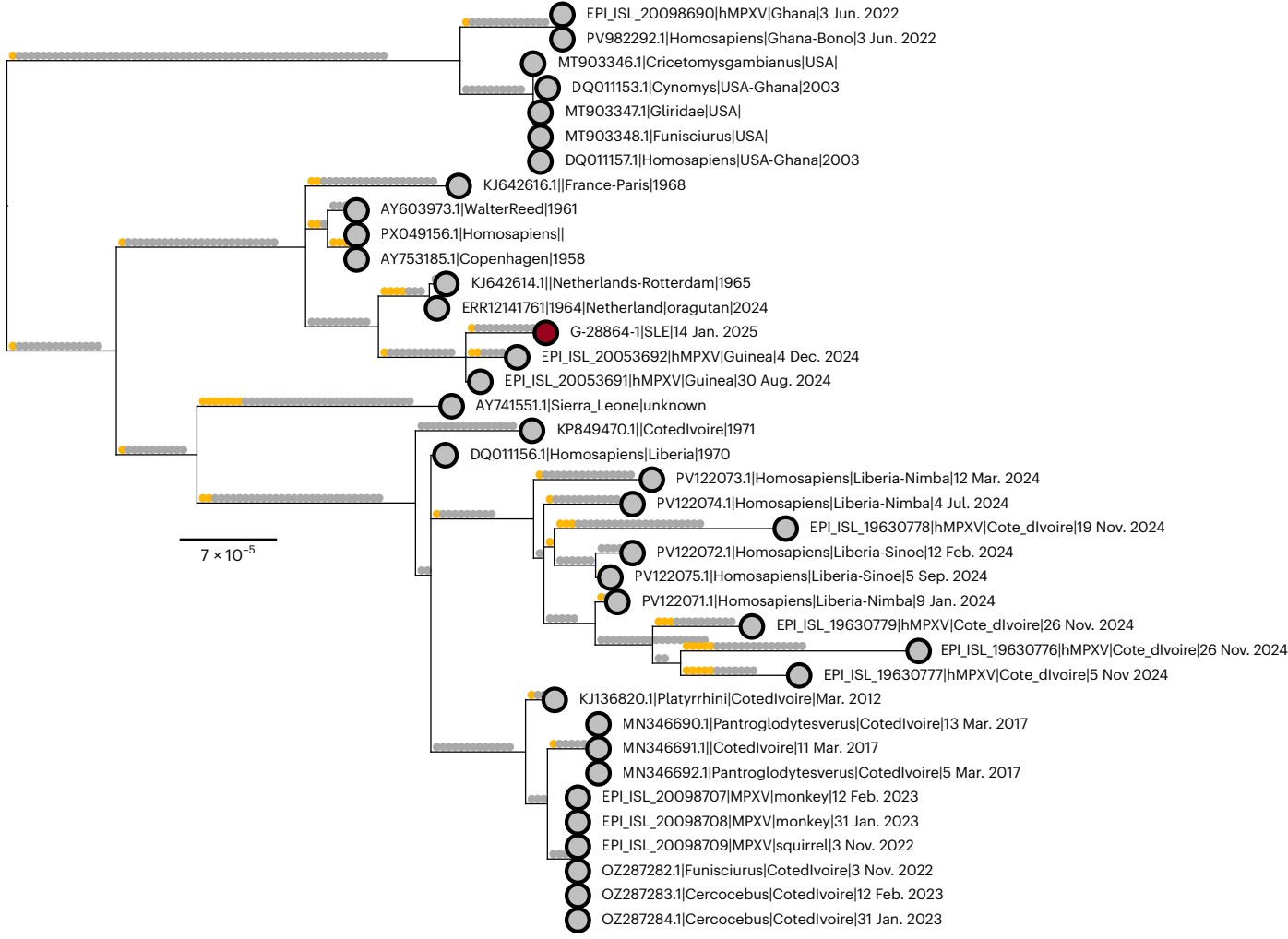

**Fig. 2 | Clade IIa phylogeny with reconstructed SNPs mapped on to branches.** The tree shows the phylogenetic placement of the new SLE clade IIa sequence (red tip, enlarged) within a cluster of recent Guinea genomes. This cluster is a sister to the historical Rotterdam or orangutan lineage. Ancestral state reconstruction was performed to map SNPs to branches. APOBEC3-characteristic substitutions (C → T or G → A in the TC or GA context) are colored yellow, whereas all other substitutions are colored gray. The tree is rooted to the new clade IIb zoonotic outgroup identified in ref. 7.

lineage A.2.2, which originated in Nigeria and continues to seed local-ized epidemics outside the country. However, due to limited sampling in the region and the divergence time inferred from APOBEC3 muta-tions along the G.1 stem branch, we cannot rule out the possibility of viral importation from an intermediate location.

Our phylogeny also indicates G.1 export from SLE (Fig. 1c). This evidence includes four sequences sampled in the USA between March and June 2025, three of which have confirmed travel histories to SLE. We also identified two sequences from Germany sampled from a sin-gle patient in March 2025 and two sequences from Guinea sampled in June 2025, with travel histories to SLE confirmed for all but the Guinea sequences. The detection of multiple viral export events indicates active local transmission in SLE at a prevalence sufficient to generate repeated international spread.

### G.1 emerged in late September 2024

The timescale inferred from accumulated APOBEC3-like mutations along the G.1 stem branch suggests that the lineage circulated unde-tected in SLE or in an unsampled external location for an extended period of time before detection. To estimate the timing of G.1's emer-gence, we used Bayesian phylogenetic reconstructions in BEAST[36,37] with a nested exponential model. Under this model, we applied an exponential growth model to the G.1 lineage while allowing the rest of

the tree to evolve under an independent exponential growth model, capturing the distinct epidemiological dynamics observed in SLE.

We estimated that the time to the most recent common ancestor (tMRCA) of the G.1 lineage was 27 September 2024 (95% highest pos-terior density (HPD) 12 August 2024 to 11 November 2024) (Fig. 2a). This tMRCA represents the lower bound when the sampled G.1 lineage became established in SLE. This suggests that the G.1 lineage may have circulated for approximately 3 months (95% HPD 2–5 months) before its detection in early January 2025.

The inferred growth rate corresponds to a doubling time of approx-imately 3 weeks (95% HPD 2.4–3.9 weeks) (Fig. 3b,c), consistent with the sharp increase in incidence observed between late April and May 2025[18]. To confirm these observations, we estimated the time-varying repro-ductive number ($R_t$) from available case counts and obtained an $R_t$ of 4.4 (95% confidence interval (CI): 3.1–5.9) in early March, consistent with both epidemiological data and the inferred doubling time (Fig. 1a)[38]. We also applied JUNIPER[39] to infer the transmission dynamics of the out-break. JUNIPER estimated an outbreak start date of 11 September 2024 (95% HPD 2 June 2025 to 11 November 2024), consistent with BEAST results, and an overall reproductive number of the outbreak, includ-ing the later decline, of 1.1 (95% HPD 1.1–1.2). In contrast, the estimated growth rate for the Nigerian epidemic, represented by the remainder of lineage A from which G.1 descended (Fig. 3a), corresponded to a

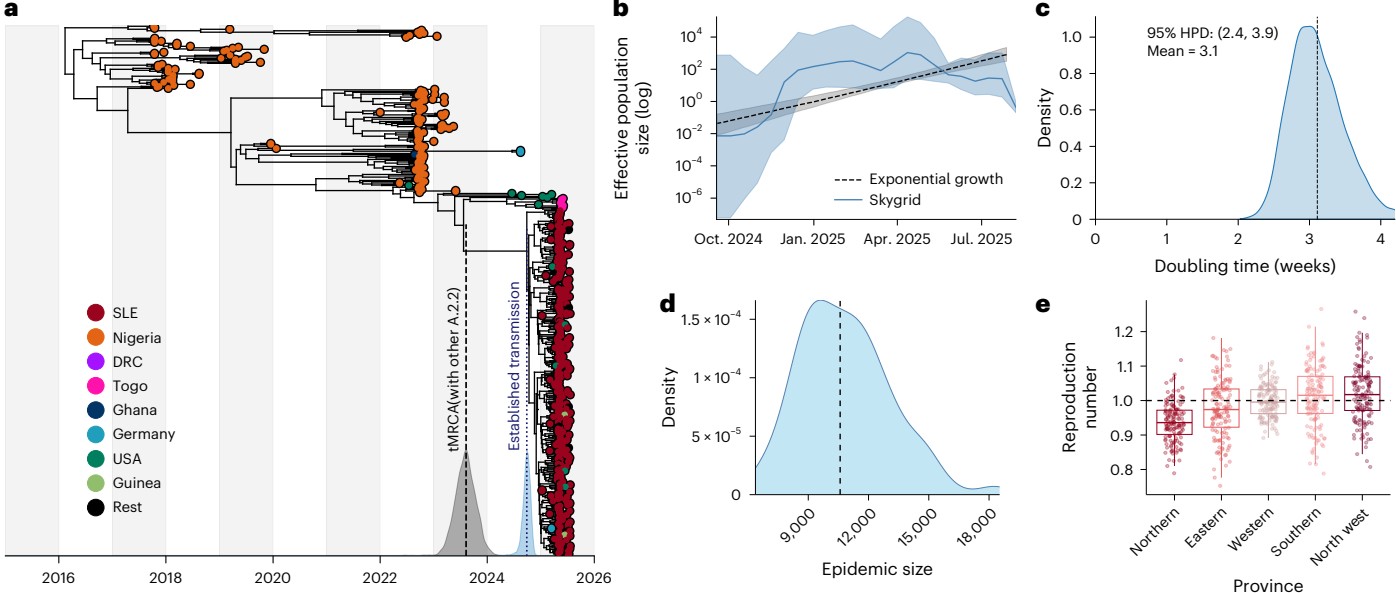

**Fig. 3 | Timing, growth dynamics and establishment of G.1 within SLE.**
**a**, Bayesian maximum clade credibility (MCC) tree of clade IIb indicating when G.1 became established in SLE. Distributions on the *x* axis show 95% HPD intervals for three time estimates: (1) the tMRCA between G.1 and its closest A.2.2 lineage (gray; estimated by BEAST), representing the upper bound of the time of introduction into SLE; (2) the tMRCA of the G.1 lineage (blue; BEAST), representing the lower bound on the time G.1 was established in SLE; and (3) the epidemic start date inferred by JUNIPER (red). Alternating gray and white vertical bands in the background each span one calendar year. **b**, Effective population size through time for the SLE epidemic under Skygrid (solid) and

exponential (dashed) coalescent models. The shaded part represents the 95% HPD interval. **c**, Posterior distribution of the estimated doubling time of the G.1 epidemic in SLE. **d**, Inferred epidemic size distribution from JUNIPER based on the estimated sampling proportion. The dashed line represents the mean. **e**, Boxplots of the reproduction number estimates per province. The points denote the average reproduction number of all sequenced cases in each province across independent Markov Chain Monte Carlo (MCMC) samples (Methods); the dashed line represents 1 (*n* = 340 sequenced genomes). The boxplot shows the median (center line), interquartile range (box bounds) and the whiskers 1.5× the interquartile range, with individual outliers shown.

doubling time of approximately 2.6 years (95% HPD 1.9–3.4 years)[7,40], indicative of a low-level endemic transmission (Extended Data Fig. 2). Our phylogenetic reconstruction under a nonparametric, Skygrid, coalescent model also supports a rapid and sustained increase in the effective population size of the G.1 lineage between April and July 2025 (Fig. 3b). We additionally used the size of clusters of identical sequences to infer a similar reproductive number (1.2, 95% CI 1.1–1.3) and an over-dispersion parameter of 0.4 (95% CI 0.2–1.0) (Extended Data Fig. 3), consistent with previous reports of mpox transmission[9].

Notably, JUNIPER estimated that the sampling proportion (the fraction of cases sequenced) was 3.3% (95% HPD 2.3–4.8%). This corresponds to an inferred total epidemic size of approximately 10,400 cases (95% HPD 7,000–15,200), compared with 5,096 confirmed cases (Fig. 3d). In addition, we estimated that the tMRCA between the G.1 lineage and the closest A.2.2.1 relative from Togo (that is, the stem branch age) was 9 August 2023 (95% HPD 3 April 2023 to 19 December 2023) (Fig. 3a). This represents an upper bound on the time of the viral introduction into SLE. However, the rapid exponential growth observed and the absence of closely related A.2.2 samples from endemic Nigeria suggest that the introduction most likely occurred shortly before the tMRCA of G.1 in late September 2024.

## G.1 was established in the WAU district
The WAU and WAR districts are the epicenter of the outbreak, accounting for approximately 80% of the mpox cases in SLE. These districts include the capital, Freetown (~1.4 million of the country's 8.6 million residents; Fig. 1b)[18]. However, it remains unclear whether the epidemic originated in this region or whether there was substantial underascertained transmission elsewhere during the early phase of the outbreak. To address these uncertainties, we performed discrete and continuous phylogeographical analyses at the district level to characterize the spatiotemporal spread of lineage G.1 within SLE.

Our discrete phylogeographical reconstructions indicate that lineage G.1 became firmly established in the WAU region, with strong posterior support (posterior probability = 0.995; Fig. 4a), consistent with the index case being reported from this district. The long stem branch implies potentially ~3 months of undetected local circulation before identification, possibly reflecting cryptic circulation in a neighboring country or unsampled transmission within SLE. We cannot confirm the site of G.1's initial emergence due to undersampling within the country and across West Africa. Nevertheless, G.1 most likely became established in WAU by late September 2024, with the district's dense, mobile population driving nationwide spread (Fig. 4a).

To account for uncertainty from sparse sampling, we applied a Markov jump-counting approach[41,42] to reconstruct the history of location changes along phylogenetic branches, allowing us to estimate the timing and origin of geographical transmission chains[41,42], defined as continuous periods of viral lineage circulation within a specific district after an introduction. The WAU was the primary source of interdistrict dissemination throughout the outbreak, with an estimated 71 introductions (95% HPD 56–88) originating from the district, equivalent to ~117 exports per million residents (HPD 92–144), far exceeding any other district (Fig. 4b).

Most early viral exports originated in the WAU region and spread into Kenema, then to Bo, Port Loko, WAR and Bonthe (Fig. 4b,c). Although WAU remained the principal source throughout the epidemic, Kenema became an important secondary hub later in the outbreak, accounting for an estimated 10 exports (95% HPD 3–21), equivalent to ~13 per million (HPD 4–27 per million; population 772,472), with exports predominantly disseminating to Bo, Pujehun, Bonthe and Falaba (Fig. 4c). The remaining districts contributed only marginally to viral dissemination. Continuous phylogeographical analysis was consistent with the discrete analyses, indicating that G.1 was first established in the WAU region before spreading early into WAR, Kenema and Bonthe (Fig. 4d).

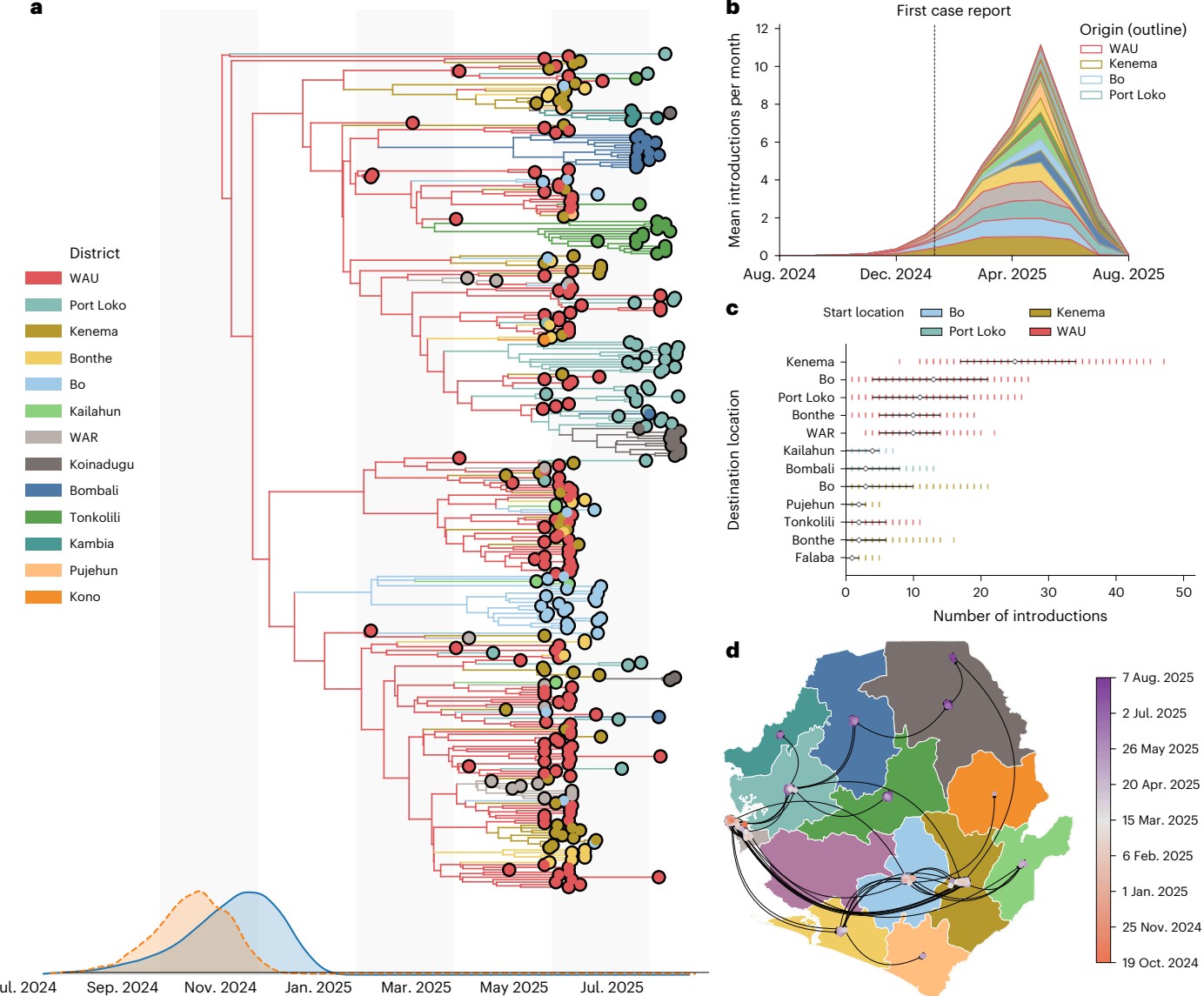

**Fig. 4 | Spatiotemporal dynamics of mpox lineage G.1 in SLE. a**, Discrete phylogeographical reconstruction showing the spatiotemporal spread of G.1 across SLE. Branches of the MCC are colored by source district, as indicated in the legend. The posterior distributions for the tMRCA of the root of the tree (orange) and for the establishment of G.1 in SLE (blue) are shown along the *x* axis. Alternating gray and white vertical bands in the background each span 2 months along the time axis. **b**, Distribution of the mean number of introductions per month by district. Each bar representing the end location of an introduction is colored by districts as in **a** and the origin district of each introduction is outlined according to the legend in **b**. **c**, Total number of introductions between districts, shown as colored bars representing the original district (*x* axis) and destination district (*y* axis), with 95% HPD intervals indicated in black (*n* = 340 genomes). The bars show 95% HPD intervals of the number of introductions estimated across 10,000 posterior phylogenetic trees. **d**, Continuous phylogeography of G.1 spatiotemporal spread across SLE, with timing of viral dissemination indicated by the color gradient as in the legend. The inferred movement of the epidemic proceeds in an anticlockwise direction. The boundary data for the map are from Global Administrative Areas (GADM) (http://www.gadm.org).

In our dataset, Port Loko was disproportionately well sequenced relative to other regions, including the WAU, with 21% of confirmed cases sequenced compared to 5% in the WAU. To account for this heterogeneity in sampling across districts, we repeated the discrete phylogeographical analysis at the regional level. The regional reconstructions were consistent with the district-level results, again indicating that G.1 most likely emerged or was first established in the western region (posterior probability = 0.99) (Extended Data Fig. 4).

Our regional analyses also supported the patterns of within-country, spatiotemporal spread observed at the district level. We found that the western region was the principal source of inter-regional viral introductions, with an estimated 52 introductions (95% HPD 39–67), equivalent to ~41 introductions per million residents

(95% HPD 31–53 per million). The earliest introductions originated in the western region and disseminated to the eastern, southern and north-west regions (Extended Data Figs. 5 and 6). As the epidemic progressed, the introduction profile shifted, with subsequent spread into other regions increasingly driven by viral exports from the western, eastern and southern regions (Extended Data Fig. 6). The northern regions contributed only marginally to interdistrict spread, with transmission there primarily seeded by introductions from the western region (Extended Data Fig. 6).

## Persistence in the WAU drove the outbreak
As shown above, repeated introductions from a limited number of districts drove the spread of lineage G.1 across SLE. However, the extent

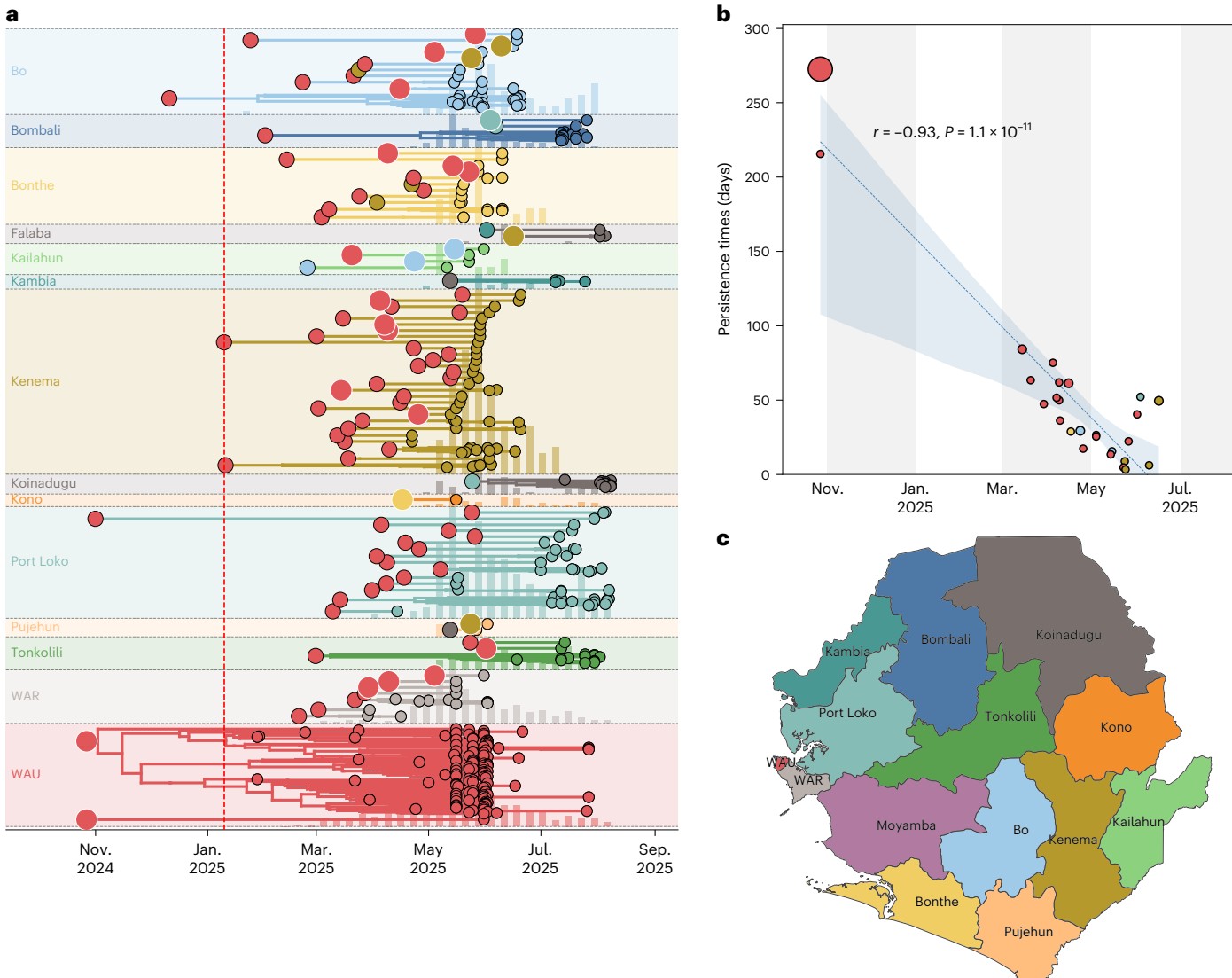

**Fig. 5 | Persistence and local transmission dynamics of G.1 across SLE.**
**a**, Duration and timing of district-level transmission chains inferred from the phylogenetic reconstruction. Each colored band represents a distinct transmission chain grouped by district (background color as in **c**). The circle at the origin of each chain denotes the district of origin, with size proportional to the number of descendant tips in that clade. Colored vertical ticks behind each band, in the matching district color, show weekly reported case counts for that district, rescaled to band height for visual context. The red dashed line marks the

date of the first reported case in SLE (10 January 2025). **b**, Relationship between clade persistence (days) and clade origin time, with points colored by district as in **a**. The fitted line shows the least-squares regression with 95% CIs. A strong negative correlation was observed (Pearson's $r = -0.93$, $P = 1.1 \times 10^{-11}$), indicating shorter transmission chain persistence later in the outbreak. Alternating gray and white vertical bands in the background each span 2 months along the time axis. **c**, Map of SLE district locations, colored consistently with **a** and **b**. The boundary data for the map are from GADM (http://www.gadm.org).

to which locally persistent transmission chains sustained ongoing transmission within districts remained unclear. To disentangle the relative contributions of local viral persistence versus new introductions, we quantified the duration of within-district viral persistence for each reconstructed transmission chain (Figs. 4a and 5a).

We found evidence of persistent within-district circulation in the WAU district from the time of emergence onward (Fig. 5a). A large transmission chain originating in this district persisted continuously throughout the sampling period (Fig. 4a). We also observed an early introduction from the WAU into Port Loko in November 2024, with the resulting transmission chain sustaining local circulation throughout the epidemic, detectable despite sparse sampling (Fig. 5a). This pattern is consistent with the index case's recent travel history to Lungi, located in the Port Loko district and home to the country's international airport[30,43]. By the time that the first cases were detected in early January 2025, active transmission chains had already been established

in four districts: WAU, Port Loko, Bo and Kenema (Fig. 4a). Several districts, including Bo, Bombali and Kenema, harbored transmission chains seeded from WAU in early 2025 that persisted locally for months (Fig. 5a,b).

From March 2025 onward, co-circulating transmission chains increased sharply, consistent with the rise in incidence between late April and May 2025 (Fig. 5a). These chains were predominantly seeded from WAU, with repeated introductions establishing multiple co-circulating but shorter-lived chains in Kenema, WAR and Bonthe. The strong negative correlation between chain duration and epidemic phase (Pearson's $r = -0.93$; Fig. 5b) reflects the impact of progressive public health interventions, including vaccination campaigns that began 26 March in WAU and WAR districts, expanded nationwide by 30 April and were complemented by ring vaccination of contacts in early June (Fig. 1a). The transition to mandatory isolation, initiated after home-based management was observed to be ineffective within this

particular setting, is credited as a key driver of the subsequent decline (Fig. 5b). Although our sampling period limits inference beyond August 2025, WAU maintained persistent transmission throughout, whereas other districts were primarily fueled by repeated introductions from this dominant hub.

## Discussion

Mpox remains an important and evolving public health threat, with both clade I and clade II now demonstrating sustained human transmission and driving large-scale outbreaks across Africa and beyond[7–12]. Since its emergence in 2014, clade IIb has persisted in human populations, fueling both the global multi-country B.1 outbreak in 2022[7,8,10,44] and now the G.1 outbreak in SLE. This shift toward sustained human-to-human transmission, observed independently across clade Ia, clade Ib and clade IIb, appears primarily driven by epidemiological factors, such as transmission in dense sexual networks, although a contribution from viral fitness cannot be ruled out. This study and others found no evidence of APOBEC3-mediated evolution that would increase transmissibility, supporting the view that recent expansions have been driven primarily by human behavioral and network factors[7,8,12,44].

Our phylogeographical analyses indicate that G.1 was most likely established in the WAU district, which served as the primary and persistent source for viral dissemination across SLE. Although neighboring countries' capitals share similar densities, none has exhibited an mpox epidemic on the scale seen in SLE. Hence, we reasoned that this specific escalation in Freetown reflects the stochastic nature of disease emergence. Population density provided the necessary conditions for amplification, but the specific introduction into high-risk networks was required to trigger an outbreak of this scale, which may explain the limited scale in nearby countries. This is corroborated by epidemiological data that showed that the early cases were concentrated within sexual transmission networks, and subsequent spread became more generalized, maintained through both household and community contact transmission alongside sexual networks—a pattern reminiscent of the recent clade Ib expansion in the DRC, Burundi and Uganda[14,18]. At the time of writing, only Guinea has reported imported cases of G.1 from the ongoing outbreak in SLE. Meanwhile, neighboring countries reported limited mpox activity during the early phase of G.1 expansion, with distinctly different lineages: Liberia reported clade IIa cases[45] and Ghana primarily reported clade IIb lineage A.2.5[46], indicative of imports from ongoing low-level transmission in Nigeria[7]. Taken together, these findings suggest that G.1's rapid expansion was driven primarily by population dynamics and network connectivity, with the extent of regional spread remaining to be clarified by future sequencing.

These findings underscore the urgent need to strengthen early warning surveillance and diagnostic access. We estimated that lineage G.1 circulated locally cryptically for approximately 3 months before detection, spreading to at least three districts beyond the WAU district during this period. However, the long stem branch connecting G.1 to its closest relative (lineage A.2.2.1 from Togo) reveals a notably deeper history: the estimated tMRCA dates to more than a year before the lineage's establishment in SLE around September 2024. This temporal gap is consistent with prolonged circulation in an unsampled location within West Africa before its introduction. Consequently, although the data suggest a regional origin, current genomic undersampling prevents us from inferring the specific country of direct introduction. The true epidemic size was estimated at 10,400 cases (5,096 laboratory confirmed), indicating that surveillance systems detected approximately half of the total cases during the outbreak. Strengthening national surveillance capacity, particularly for early detection, will require proactive, community-based case finding, decentralized diagnostics with stable supply chains and real-time genomic sequencing.

At the same time, strengthening frontline capacity is essential to translate surveillance data into effective outbreak control. Training of healthcare workers and community health volunteers should emphasize the recognition of mpox's early and distinguishing signs and its differentiation from varicella, both of which are endemic[47–49]. It should reinforce adherence to standardized case definitions, safe specimen collection and rigorous infection prevention and control measures, including isolation, hand hygiene, the use of personal protective equipment and safe caregiving. In addition, risk-reduction counseling and practical guidance are needed to help communities limit the spread in households, schools, markets and healthcare facilities. These investments will improve detection and case management not only for mpox but also for other epidemic-prone diseases[43].

Mpox incidence in SLE peaked during weeks 18–19 (late April to mid-May), then declined as coordinated response measures took effect. By 17 August 2025, 147,779 individuals had been vaccinated (~1.96% national coverage), including healthcare workers (23%), identified contacts (43%), high-risk groups (12%), which include sex workers, MSM individuals, household contacts of confirmed cases and people living with HIV and others (22%); 51% were male and 83% aged 20–49 years. Case decline aligns closely with successive intervention milestones: vaccination in WAU and WAR districts (week 13), the nationwide campaign (week 18), immediate vaccination of high-risk contacts (week 20), ring vaccination (week 23) and the enhanced integrated mpox response (week 26). This enhanced phase was initiated after the Ministry of Health observed that home-based management was inadvertently sustaining transmission networks. The subsequent shift to mandatory isolation is credited as a primary driver of the rapid decline and return to zero cases, contrasting with the lingering low-level circulation seen in Nigeria[7,22] and the DRC, where zoonotic transmission continues to reseed human infections[7,12,22]. Together, these interventions underscore the importance of layered, timely vaccination and effective case isolation, supported by robust surveillance, for controlling mpox transmission.

This study has several limitations, primarily related to the sample's coverage and spatial heterogeneity. Although our dataset includes ~6.9% of confirmed cases, coverage varied widely across districts, ranging from fewer than 5% of cases in the high-burden WAU (131 of 2,867) and WAR (15 of 1,020) districts to more than 30% in Kenema, Bo and Bonthe, potentially overestimating exports or introduction from well-sampled districts. To assess robustness, we aggregated districts into broader regions and tested for sampling artifacts using a generalized linear model. These analyses consistently supported the western region as the emergence location and rejected sampling intensity as a driver of diffusion. Nevertheless, lower coverage outside the capital means that our estimates of local persistence are conservative lower bounds and shorter transmission chains in these districts may reflect undersampling rather than true extinction. Reproductive number estimates are further constrained by underascertainment, with the true epidemic size nearly doubling official case counts. Although the WAU district served as the primary source for amplification and dissemination, we cannot definitively exclude cryptic establishment in an unsampled district or neighboring country before the outbreak's explosive detection in the capital.

Nevertheless, the primary phylogeographical inferences, WAU as the source of spread, with early seeding of Kenema, Bo, Port Loko, WAR and Bonthe, are well supported by independent epidemiological patterns in the case data. The lack of detailed clinical and behavioral metadata limits direct linkage of genomic patterns to disease severity or transmission context. This study demonstrates how local genomic capacity, when paired with timely data, can transform outbreak response from reactive to informed. Yet, without equitable access to diagnostics, vaccines and therapeutics, mpox and other zoonotic pathogens will continue to emerge and sustain transmission across Africa[50]. Expanding real-time sequencing, representative surveillance and integrated clinical–genomic data systems will be critical to ensuring that early detection translates into effective control, not only for mpox but for future emerging pathogens.

## Online content

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

Allan K. O. Campbell [1,2,3,27]✉, John Demby Sandi [3,4,5,27]✉, Ifeanyi F. Omah [6,7], Martin Faye [8], Edyth Parker [9], Taylor Brock-Fisher [10], Crystal M. Gigante [11], Vidalyn Folorunso [2], Mohamed Saio Kamara [4], Anu Jegede Williams [2], Mouhamed Kane [8], Tiangay Mariama Patience Sallay Kallon [4], Julian Campbell [2,3], Kadiatu Salmata Sesay [4], Sia Mani [2], Choe Miller [2], Naomi Daniel Sesay [2], Francis Baimba [4], Mignane Ndiaye [8], Roberta Lansana [2], Ibrahim Umaru Fofanah [4], Simon Ruhweza [1], Zein Souma [2], Amanda Kargbo [2], Komba Koninga [2], Alie Tia [12], Jone Ngobeh [12], Foday Thoronka [12], Alusine Fofanah [1,13], Al Ozonoff [10], Colby Wilkason [10], Danny Park [10], Christopher Tomkins-Tinch [10], Marietou F. Paye [10], Christopher Shin [10], Ian Baudi [10], Brendan Blumenstiel [10], Patrick Varilly [10], Ivan Specht [10,14], Ben Fry [15], Karlie Zhao [15], Paul Cronan [15], Ellory Laning [15], Oludayo Oluwaseyi Ope-ewe [9], Ayotunde Elijah Sijuwola [9], Femi Saibu [9], Harouna Soumare [9], Ebenezer Kehinde Ogundana [9,16], Ruth Oluwaseun Obaado [9], Jolly Amoche Adole [9], Imonikhe Kennedy Kio [9], Folefac Agnes Njeandoh [9], Alexandra Tuttle [11], Walter O. Oguta [17], Jonathan Greene [18], Aminata Koroma [1], Joseph Kanu [1], Mamadou Aliou Barry [8], Aboubacry Gaye [8], Andy Mahine Diouf [8], Christine Hughes [11], Joshua I. Levy [19], Alhaji N'jai [20,21,22], Moussa Moise Diagne [8], Dolo Nosamiefan [9], George Ameh [18], John Klena [11], Monique A. Foster [11], Abebaw Kebede [23], Collins Tanui [23], Boubacar Diallo [8], Anise Happi [9], Sofonias Tessema [23], Abdourahmane Sow [8], Yenew Kebede [23], Isatta Wurie [3], James Squire [1], Doris Harding [1], Zikan Koroma [1], Mohamed Boie Jalloh [1,24], Amadou Alpha Sall [8], Kristian G. Andersen [19], Andrew Rambaut [6], Mohamed Alex Vandi [1], Ibrahima Socé Fall [8], Pardis Sabeti [10,25], Christian Happi [9,16,25], Foday Sahr [1] & Donald S. Grant [1,24,26]✉

[1]National Public Health Agency, Freetown, Sierra Leone. [2]Central Public Health Reference Laboratory, Freetown, Sierra Leone. [3]Faculty of Medical Laboratory Sciences and Diagnostics, College of Medicine and Allied Health Sciences, University of Sierra Leone, Freetown, Sierra Leone. [4]Kenema Government Hospital, Kenema, Sierra Leone. [5]School of Community Health Sciences, College of Medical Sciences, Njala University, Bo, Sierra Leone. [6]Institute of Ecology and Evolution, University of Edinburgh, Edinburgh, UK. [7]Department of Parasitology and Entomology, Nnamdi Azikiwe University, Awka, Nigeria. [8]Institut Pasteur de Dakar, Dakar, Senegal. [9]Institute of Genomics and Global Health, Redeemer's University, Ede, Nigeria. [10]The Broad Institute of MIT and Harvard, Cambridge, MA, USA. [11]Centers for Disease Control and Prevention (CDC), Atlanta, GA, USA. [12]Sierra Leone–China P3 Friendship Laboratory, Freetown, Sierra Leone. [13]School of Public Health, Njala University, Bo, Sierra Leone. [14]Institute for Computational and Mathematical Engineering, Stanford University, Stanford, CA, USA. [15]Fathom Information Design, Boston, MA, USA. [16]Department of Biological Sciences, Faculty of Natural Sciences, Redeemer's University, Ede, Nigeria. [17]World Health Organization Regional Office for Africa (WHO AFRO), Dakar, Senegal. [18]World Health Organization (WHO), Freetown, Sierra Leone. [19]Department of Translational Medicine, The Scripps Research Institute, La Jolla, CA, USA. [20]Department of Biological Sciences, Fourah Bay College, University of Sierra Leone, Freetown, Sierra Leone. [21]Department of Microbiology, College of Medicine and Allied Health Sciences, University of Sierra Leone, Freetown, Sierra Leone. [22]Department of Medical Education, California University of Science and Medicine, Colton, CA, USA. [23]Africa Centres for Disease Control and Prevention (Africa CDC), Addis Ababa, Ethiopia. [24]College of Medicine and Allied Health Sciences, University of Sierra Leone, Freetown, Sierra Leone. [25]Department of Immunology and Infectious Diseases, Harvard T.H. Chan School of Public Health, Harvard University, Boston, MA, USA. [26]Department of Clinical Sciences, Liverpool School of Tropical Medicine, Liverpool, UK. [27]These authors contributed equally: Allan K. O. Campbell, John Demby Sandi. ✉e-mail: oakcampbell25@gmail.com; johnatsandi@gmail.com; donkumfel@yahoo.co.uk

## Methods

### Ethics and inclusion statement

This work was conducted as part of the public health response to the mpox outbreak under the mandate of the National Public Health Agency, Ministry of Health, Sierra Leone and all activities were performed in accordance with national outbreak investigation guidelines. As this investigation formed part of an emergency public health response, it was exempt from institutional ethics review and the requirement for individual informed consent under national public health regulations. All data were anonymized before analysis to protect patient confidentiality. All authors met established authorship criteria and contributed substantially to study design, data generation, analysis, interpretation and paper preparation. Local public health scientists and laboratory personnel were actively involved throughout the investigation and analysis, ensuring equitable collaboration, local ownership and appropriate recognition of contributions. Care was taken to minimize stigma and harm in data handling, analysis and reporting.

### Sample collection and molecular testing

All samples were collected between 10 January and 3 August 2025 from outbreak response surveillance. The dataset was produced through collaborative efforts among the CPHRL, KGH VHF Lab and the IPD mobile laboratory in Port Loko. Samples were processed for DNA extraction at CPHRL and KGH VHF Lab, using the RADI FAST DNA Extraction Kit (KH Medical), following the manufacturer's standardized protocols. The extracted DNA was then tested for the presence of MPXV using the RADI FAST mpox detection kit via real-time quantitative PCR, following the manufacturer's instructions. At the IPD or Port Loko lab, DNA extraction was performed using the QIAamp DNA mini-kit (50), following the manufacturer's instructions (QIAGEN) and quantitative PCR was done using the lyophilized 1-step RT-PCR Polymerase Mix kit with the Lightmix modular Dx monkeypox virus assay, according to the manufacturer's instructions (Roche, Tibmol Biol). Every experiment included appropriate internal and external quality controls.

### Next-generation sequencing

We prioritized mpox-positive samples with a cycle threshold ($C_t$) value ≤30 for sequencing. Libraries were prepared using the Illumina RNA Prep with Enrichment (L) Kit and enriched with the Viral Surveillance Panel 2.0 targeting epidemic-prone pathogens, including MPXV. After capture with streptavidin-coated magnetic beads, libraries were amplified, purified, quantified and quality checked using Qubit (Thermo Fisher Scientific) and Agilent TapeStation (Agilent Technologies). Samples from CPHRL were sequenced on the Illumina MiniSeq with 2× 150-bp paired-end reads. Sequences from KGH were generated using the Illumina Viral Surveillance Panel, followed by Illumina sequencing on Illumina Miseq. Sequences at the IPD mobile lab were generated using an amplicon-based method with specific mpox primer pools, followed by Oxford Nanopore Technologies MK1B or Illumina iSeq100 sequencing[51].

### Genome assembly

The CPHRL sequences were generated with an in-house reference-based assembly pipeline. Briefly, we mapped reads against a clade IIb reference genome (NC_063383, an early clade IIb/sh2017 genome from Nigeria) with BWA-MEM[52] and called consensus using samtools[53] and iVar[54]. Sequences from KGH were assembled using the Broad Institute viral-ngs assemble_ref_based pipeline implemented in Terra using a clade IIb reference genome (NC_063383). Sequences from IPD were assembled from Oxford Nanopore Technologies data, using an in-house, reference-based assembly pipeline with a clade IIb reference genome (NC_063383) and a minimum read depth of 100. For Ilumina data, the trimmed BAM files were mapped to a clade IIb reference genome (NC_063383) using BWA-MEM (v0.7.17-r1188)[52]. The generated BAM mapping files were sorted using SAMtools (v1.6)[53] and then used as input to iVar (v1.3.1)[54] with a minimum read depth of 10 and depth threshold of 0.5, to generate consensus sequences. The mean coverage across all sequences ranged from 25× to 8,100×, with 324 of the 338 genomes exceeding 70% completeness.

In total, we generated 338 high-quality sequences. The total sequences produced were from: CPHRL ($n = 128$), KGH ($n = 105$) and the IPD mobile laboratory in Port Loko ($n = 105$). Our dataset included sequences sampled between 10 January and 3 August 2025 from 14 districts: WAU ($n = 129$), Kenema (46), Port Loko (39), Bo (32), Bonthe (18), Bombali (16), Tonkolili (16), Kailahun (5), WAR (15), Kambia (4), Koinadugu (12), Falaba (3), Kono (1) and Pujehun (2).

### Genomic dataset curation

We combined our 338 genomes with all high-quality (genome coverage >70%), publicly available clade IIb MPXV genomes from Pathoplexus[55]. Only one representative of the multi-country outbreak lineage B.1 was included, because it was not the primary focus of our study. We also included the closest zoonotic outgroup to clade IIb (PP852949.1) sampled in 2022 from Nigeria, which was used to root the tree[7]. In total, the dataset comprises 530 sequences.

### Epidemiological data

We obtained the time series of confirmed mpox cases from the National Public Health Agency. We used EpiFilter or Smooth to infer a time-varying $R_t$ (ref. 38).

### Phylogenetic analysis

We aligned our dataset to the clade IIb reference genome (NC_063383) using the squirrel package developed in refs. 8,56. The alignment was trimmed and the 3′-terminal repeat regions, as well as repetitive regions, regions with low complexity, clustered mutations or mutations near gaps or ambiguities were masked using the package's quality control mode.

The full MPXV phylogeny was reconstructed with IQ-TREE v2.0 implemented within the squirrel package[57]. We reconstructed a separate clade IIb phylogeny under the default parameters in Squirrel and rooted it to the zoonotic outgroup PP852949 from Nigeria in 2022[7], which was subsequently pruned. All zero-length branches were collapsed and an ancestral state reconstruction was performed across the clade IIb phylogeny using IQ-TREE2[57]. Lineages were assigned from the nomenclature established in refs. 3,4 using the Nextclade tool[58].

### Bayesian phylodynamics

To estimate the timing of G.1's emergence, we adopted the partitioned model developed in ref. 8 to model APOBEC3-mediated evolution implemented in the BEAST X v10.5.0 software package[36,37] with the BEAGLE high-performance computing library[59]. We used a nested exponential coalescent model to adequately model G.1's distinct epidemiological dynamics. We applied an exponential growth model to the tree from the most recent common ancestor of the G.1 lineage onward. The rest of the tree, representing the remainder of lineage A that is the source population, was modeled under an independent exponential growth model. This allows us to estimate distinct growth rates and associated doubling times for the epidemic in SLE and the wider West Africa (predominantly Nigeria), respectively. We ran two independent chains of 100 million states to ensure convergence, discarding the initial 10% of each chain as burn-in. The chains were then combined using LogCombiner. For all subsequent analyses, we assessed convergence using Tracer and constructed an MCC tree in TreeAnnotator 1.10[60].

### Phylogeographic analysis

To investigate the spatial spread of the G.1 lineage within SLE, we reconstructed the timing and pattern of geographical transitions between SLE's districts and at the regional level under an asymmetrical discrete trait analysis[41]. We applied Bayesian stochastic search variable selection

to identify statistically supported migration routes. We used a Skygrid coalescent model with 34 change points distributed over 7 months to capture the doubling time of 2-week intervals, excluding lineage A sequences from other countries[61]. We combined two independent MCMC chains of 100 million states sampled every 10,000 states, discarding the initial 10% of trees as burn-in. We confirmed that all effective sample size (ESS) values were >200.

We employed a Markov jump-counting procedure across the full posterior distribution to further investigate the timing and origin of geographical transitions while accounting for uncertainty in our phylogeographical reconstructions. We used the TreeMarkovJumpHistoryAnalyzer implemented in BEAST X[36,37] to extract all Markov jumps from our posterior tree distributions[62]. We accounted for the uneven distribution of sequences across SLE districts by performing all phylogeographical analyses on an aggregated, regional level.

We also performed a continuous phylogeographical analysis to quantify the dispersal of the G.1 lineage across regions and districts in SLE. We assigned each sequence the latitude and longitude of the centroid of its sampling district. We used the Skygrid coalescent model described above, with a Cauchy distribution to model the among-branch heterogeneity in dispersal velocity[42]. We ran two independent MCMC chains of 50 million states, sampling every 2,000 states. We combined the chains after discarding 10% of the states as burn-in.

## Transmission reconstruction

We used the R package JUNIPER[39] to conduct outbreak reconstruction from all clade IIb G.1 samples based on pathogen genomes. JUNIPER infers transmission networks using both sequenced and unsequenced cases. We masked our clade IIb alignment only to include the APOBEC3 target site partition. Specifically, as APOBEC3 proteins mutate TC dimers to TT dimers and GA dimers to AA dimers, we retained only pairs of positions on the aligned genomes that exhibited TC or TT in all sequences or GA or AA in all sequences. We removed the conserved nucleotide in each dimer across all sequences, to retain only the second nucleotide of the TC or TT dimers, and retained only the first nucleotide of the GA or AA dimers. A substitution rate of $6.9 \times 10^{-7}$ was calculated by assuming 6 APOBEC3-like mutations per site per year across the partitioned genome length of 23,602 (refs. [7,8]). JUNIPER was run for a total of 30,000 iterations (across two 15,000 iteration runs) with a fixed mutation rate, a generation interval of 11.4 d and a sojourn interval of 15 d. After completion of the runs, 20% burn-in was removed and the resulting chains were downsampled to an effective sample size of 69 and 73, respectively, to ensure minimal autocorrelation between samples[63] and combined for a total ESS of 142.

The estimated outbreak size and estimated outbreak start date were directly inferred from the results. To calculate the per-individual reproductive number, we computed the expected number of total transmissions under the stochastic–epidemic model. The conditional probability $\mathbb{P}_t(X = k \mid d)$ that an individual infected at time $t$ transmits $k$ total people, given that $d$ offspring are explicitly represented in the transmission network inferred by JUNIPER, is

$$\mathbb{P}_t(X = k \mid d) = \frac{1}{Z} \binom{k}{d} f(k) w_t^{k-d}$$

where $\omega_t$ is the probability that a case infected at time $t$ has no sequenced descendants up through the time of last sample collection, $f(k)$ is the probability of one person transmitting to $k$ other people, and

$$Z = \sum_{k=d}^{\infty} \mathbb{P}_t(X = k \mid d)$$

The equation is adapted from TransPhylo[64]. The expected number of transmissions for said individual, conditional on the $d$ offspring in the transmission network inferred by JUNIPER, is then given by

$$\mathbb{E}_t[X \mid d] = \sum_{k=d}^{\infty} k \mathbb{P}_t(X = k \mid d).$$

Reproductive numbers within any metadata category were computed by averaging the expected number of transmissions for each observed individual in said category.

## Statistical analysis

Sample selection and quality control sample size were determined by cases meeting sequencing thresholds ($C_t \leq 30$, genome coverage >60% at ≥25× depth) during the outbreak (10 January to 3 August 2025). We included 338 high-quality MPXV genomes, representing 6.9% of total PCR-confirmed cases ($n = 5,328$). No samples were excluded post-hoc. Phylogenetic and mutational analysis phylogenetic confidence was assessed via ultrafast bootstrap (UFBoot) in IQ-TREE v2.0 (1,000 replicates); support ≥70% was considered significant. Bayesian phylodynamic parameters were estimated using BEAST X v10.5.0. MCMC chains (two independent runs) were executed for 100 million states (sampling every 10,000) for general analyses and 50 million states (sampling every 2,000) for phylogeographical analyses. We applied a 10% burn-in and confirmed convergence via Tracer v1.7 (ESS > 200). All parameters are reported as posterior means with 95% HPD intervals. For transmission and reproduction modeling, time-varying effective reproduction numbers were derived from daily case counts using EpiFilter v1.0 (Smooth mode) with a serial interval of 11.4 d. For outbreak reconstruction, we used JUNIPER (v1.0) to jointly infer transmission networks from sequenced and unsequenced cases (30,000 MCMC iterations; fixed rate $6.9 \times 10^{-7}$ substitutions per site per year). Total epidemic size and sampling proportions were inferred within this framework. For overdispersion, individual-level and overdispersion were estimated via a negative binomial model of identical sequence clusters and used to identify superspreading dynamics. Phylogeography and persistence discrete phylogeography employed an asymmetrical continuous time Markov chain model with Bayesian stochastic search variable selection; migration routes were supported at a Bayes factor >3. Interdistrict introductions were quantified using Markov jump counting across 10,000 posterior trees. Chain persistence was defined as the duration between inferred origin and the most recent sampling date. The association between origin time and persistence was assessed using a two-tailed Pearson's correlation. Software and reproducibility analyses and visualization were performed in R (v4.3.1), Python and Baltic. All code is available at GitHub (https://github.com/Ifeanyi-omah/Sierra_Leone_Mpox_project).

## Reporting summary

Further information on research design is available in the Nature Portfolio Reporting Summary linked to this article.

## Data availability

All 338 MPXV genome sequences generated in this study are deposited on Pathoplexus (https://doi.org/10.62599/PP_SS_232.1). Epidemiological case count data used in this study are publicly available from the National Public Health Agency of Sierra Leone at https://clt.npha.gov.sl/outbreak.aspx. The data underlying each main figure, including the case count time series, BEAST log files and phylogeny files, are available on GitHub (https://github.com/Ifeanyi-omah/Sierra_Leone_Mpox_project). No other primary data were generated in this study.

## Code availability

All code used to run the phylogenetic, phylodynamic, phylogeographical and transmission reconstruction analyses described in this study is publicly available on GitHub (https://github.com/Ifeanyi-omah/Sierra_Leone_Mpox_project). The repository includes all source data underlying each main figure, including the case count time series,

BEAST log files, maximum clade credibility trees and phylogeographical output files. No customized code was developed that is not available in this repository.

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

## Acknowledgements

The IPD acknowledges M. Cissé (data scientist) and M. Diarra (laboratory technician), who were deployed with the IPD team in SLE, as well as N. Dia, O. Faye, C. Loucoubar, D. Yero Sow and O. Faye for logistics and coordination support. IPD also thanks L. Boussiengui and F. Diène Thiaw for generating the first sequences from SLE at IPD. The US CDC acknowledges members of the poxvirus laboratory who contributed to sequencing of US samples included in this study, as well as staff from the Tennessee Department of Health, Maryland State Department of Health, Minnesota Department of Health, Texas Department of State Health Services, Pennsylvania Department of Health and Virginia Department of Health for submitting samples for MPXV genomic surveillance. The funders had no role in study design, data collection and analysis, decision to publish or preparation of the manuscript. Africa CDC, through the Africa Pathogen Genomics Initiative, supported the establishment of in-country sequencing capacity at CPHRL through the provision of sequencing equipment, training, testing and sequencing reagents and the World Health Organization also provided critical support through the provision of sample storage equipment and diagnostic and sequencing reagent supplies, complemented by essential training at the national level. This work was also made possible by support from Flu Lab and a cohort of generous donors through TED's Audacious Project, including the ELMA Foundation, M. Scott, the Skoll Foundation and Open Philanthropy. The Institute of Genomics and Global Health team also received support from the Rockefeller Foundation (award no. 2021 HTH 017). This work of the IPD team in terms of sample testing, sequencing, capacity building, data analysis and dashboarding and coordination was funded by West African Health Organization (WAHO/OOAS), in support of the SLE national mpox response efforts. I.F.O. was supported by the Wellcome Trust Hosts, Pathogens & Global Health program (Wellcome Trust, grant no. 218471/Z/19/Z) in partnership with Tackling Infectious Disease to Benefit Africa. A.R. acknowledges the support of the Wellcome Trust through the ARTIC Network (award nos. 206298/Z/17/Z and 313694/Z/24/Z). The contents of this manuscript are solely the responsibility of the authors and do not necessarily represent the official views of the US CDC or the US Department of Health and Human Services.

## Author contributions

A.K.O.C. and J.D.S. conceived the study. A.K.O.C., J.D.S., I.F.O., E.P., M.F., A.R., K.G.A. and P.S. developed the methodology. Investigation was performed by A.K.O.C., J.D.S., M.S.K., K.S.S., F.B., I.U.F., A.K., K.K., J.N., F.T., A.F., M.B.J., F.S., J.S., D.H., Z.K., M.A.V. and I.S.F. and additional contributors from collaborating institutions. Data curation was carried out by A.K.O.C., J.D.S., V.F., A.J.W., J.C., S.M., C.M., N.D.S., R.L., Z.S., A.K., K.K., A.T. and C.M.G. Formal analysis was conducted by I.F.O., E.P., M.F., A.R., K.G.A., P.S., C.T.-T., D.P., B.B., P.V., I.S., C.W., T.B.-F., I.B., C.S. and M.F.P., with software support provided by I.F.O., E.P., A.R., K.G.A., C.T.-T., D.P., B.B., P.V., I.S., C.W., T.B.-F., I.B. and C.S. Validation was performed by A.K.O.C., J.D.S., I.F.O., E.P., M.F., A.R. and K.G.A. and visualization by I.F.O., E.P., M.F., A.R., K.G.A. and P.S. Resources were provided by A.K.O.C., J.D.S., M.S.K., A.K., J.K., W.O.O., J.G., G.A., M.A.F., A.T. and institutional partners. Project administration was undertaken by A.K.O.C., J.D.S., M.F. and D.S.G. Supervision was provided by D.S.G., C.H., P.S., K.G.A., A.A.S. and A.R. Funding acquisition was done by D.S.G., C.H., P.S., K.G.A. and A.A.S. A.K.O.C., J.D.S., I.F.O., E.P., M.F. and A.R. contributed to the original draft of the paper and all authors contributed to review and editing and approved the final version.

## Competing interests

The authors declare no competing interest.

## Additional information

**Extended data** is available for this paper at https://doi.org/10.1038/s41591-026-04385-8.

**Correspondence and requests for materials** should be addressed to Allan K. O. Campbell, John Demby Sandi or Donald S. Grant.

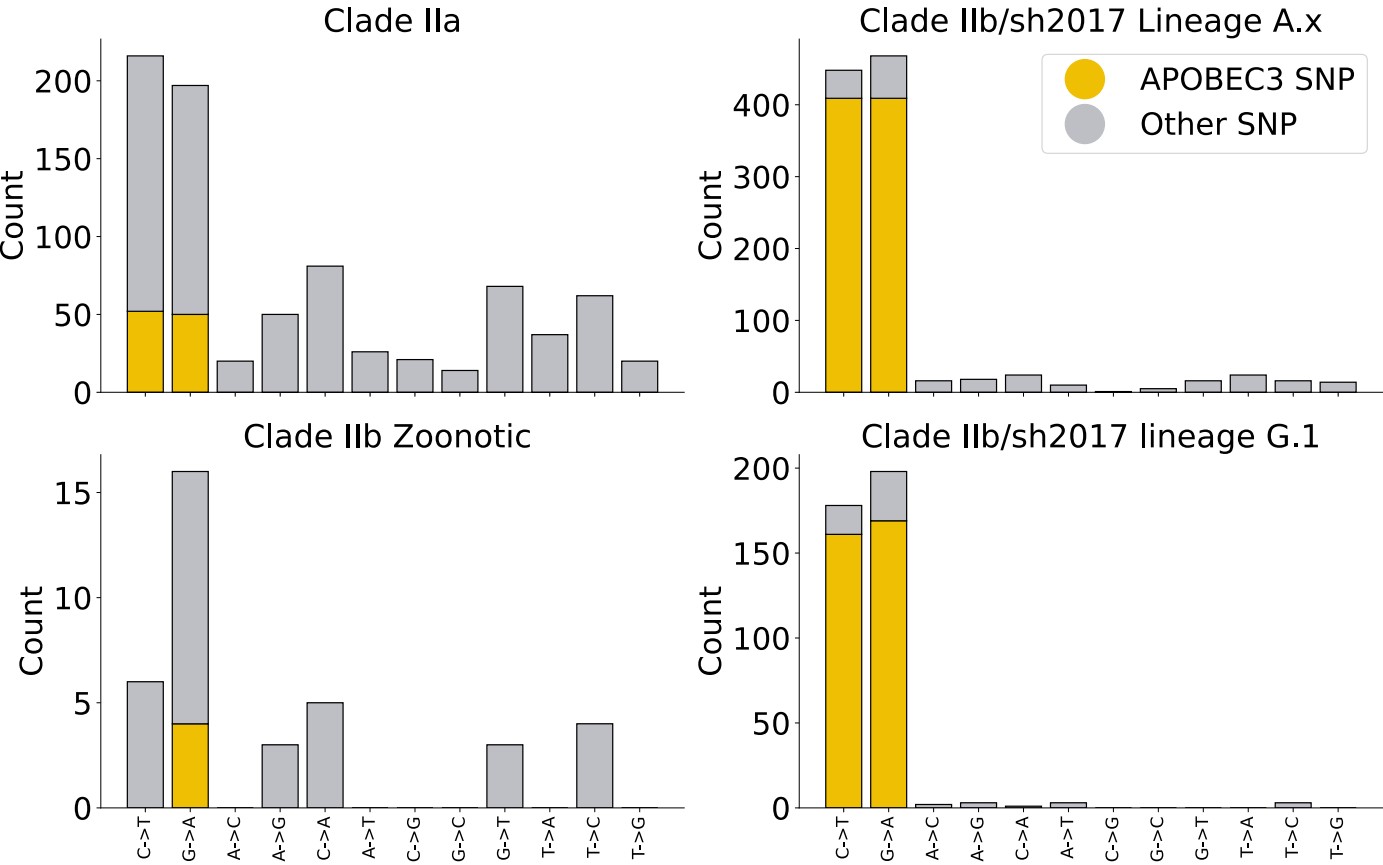

**Extended Data Fig. 1 | Comparative mutational spectra across MPXV lineages.** Yellow bars indicate APOBEC3-signature mutations (C → T and G → A in TC/GA context), while grey bars represent all other substitutions. The massive enrichment in Lineage A.x and G.1 contrasts with the background mutation patterns seen in zoonotic lineages.

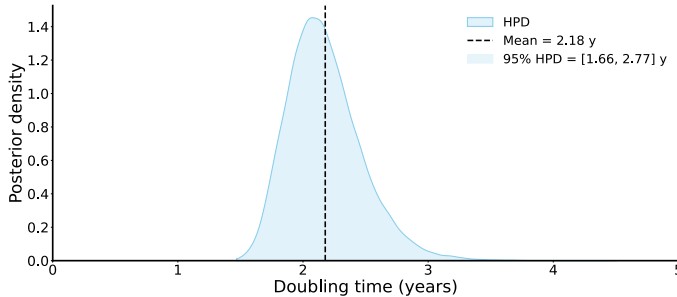

**Extended Data Fig. 2 | The doubling time of the Nigeria Epidemic.** The distribution of the doubling time for the Nigerian epidemic from 2017 to 2022.

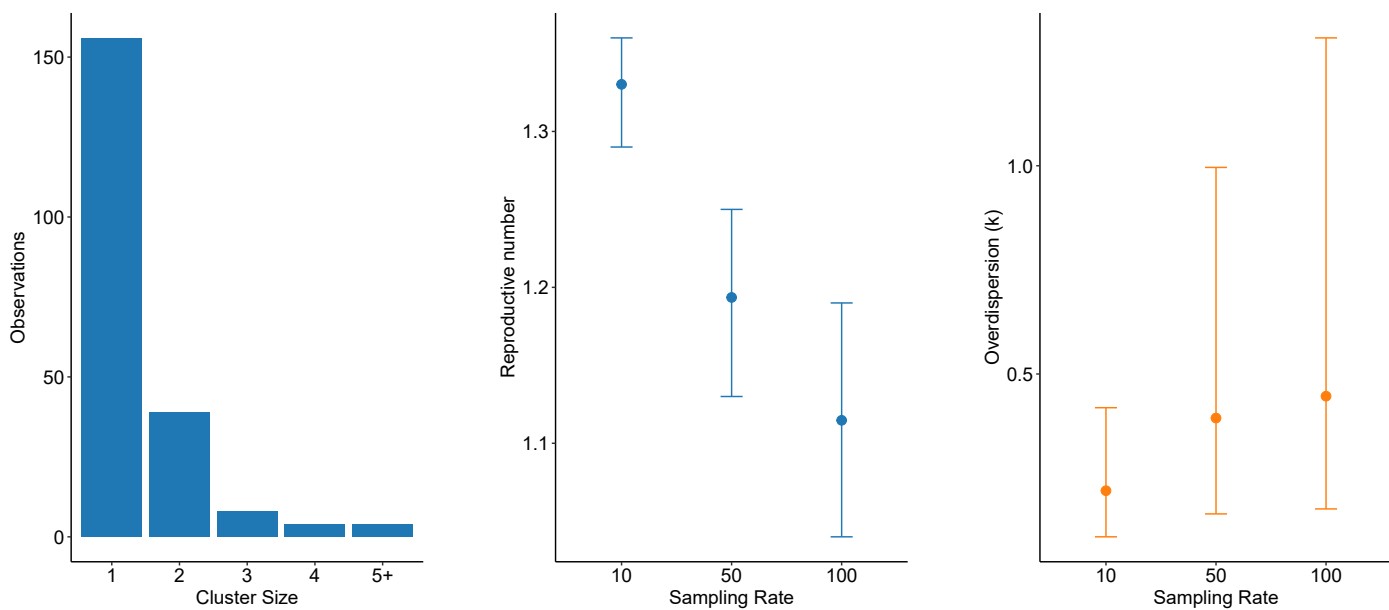

**Extended Data Fig. 3 | Distribution of the size of clusters of identical sequences during the mpox outbreak.** Estimates for the reproduction number R and the dispersion parameter k assuming that 10, 50 and 100% were detected as cases.

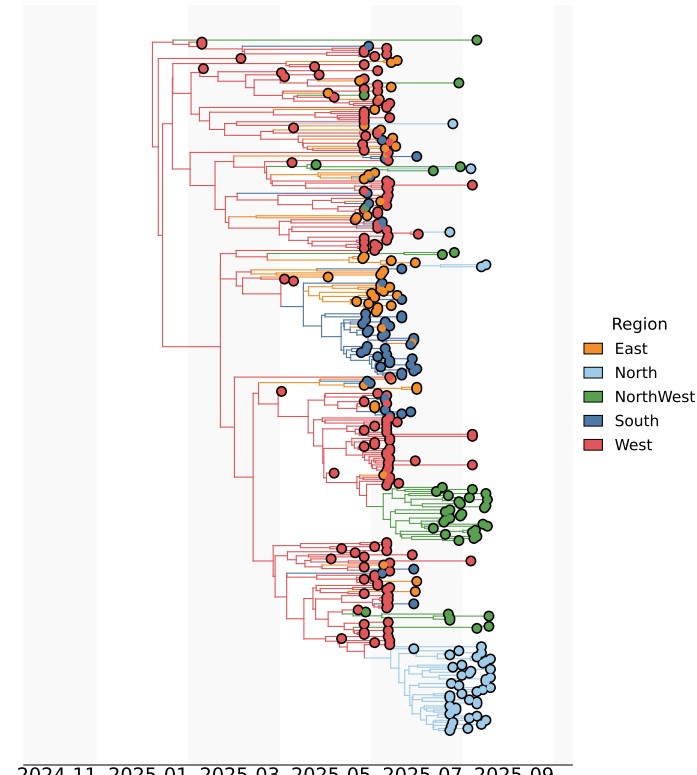

2024-11  2025-01  2025-03  2025-05  2025-07  2025-09

**Extended Data Fig. 4 | Phylogeographic analyses of the G.1 lineage in Sierra Leone on a regional level.** The branches of the Maximum Clade Credibility tree (MCC) are coloured by region, as per legend.

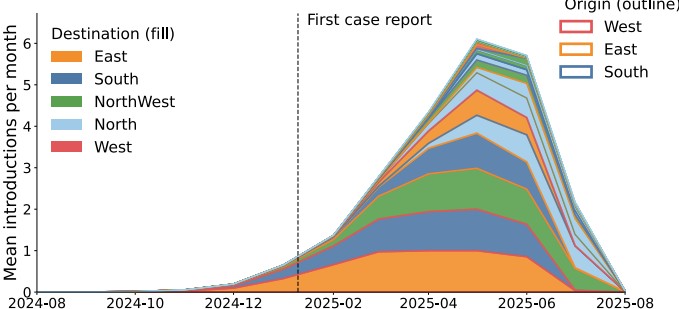

**Extended Data Fig. 5 | The distribution of the number of introductions across time by region. Data are summarised across the posterior of 10,000 trees.** Distribution represents the 95% highest posterior density. The end location state is coloured by region, as per legend. The start location is highlighted by an outline of the area.

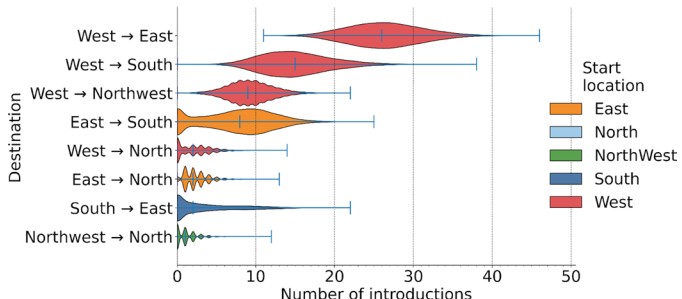

**Extended Data Fig. 6 | Total number of introductions by region from each start location.** Data are summarised across the posterior, annotated as per the legend, and colour-coded by end location on the y-axis.

# Reporting Summary

## Statistics

For all statistical analyses, confirm that the following items are present in the figure legend, table legend, main text, or Methods section.

| n/a | Confirmed | |
|---|---|---|
| ☐ | ☒ | The exact sample size (*n*) for each experimental group/condition, given as a discrete number and unit of measurement |
| ☐ | ☒ | A statement on whether measurements were taken from distinct samples or whether the same sample was measured repeatedly |
| ☒ | ☐ | The statistical test(s) used AND whether they are one- or two-sided *Only common tests should be described solely by name; describe more complex techniques in the Methods section.* |
| ☐ | ☒ | A description of all covariates tested |
| ☒ | ☐ | A description of any assumptions or corrections, such as tests of normality and adjustment for multiple comparisons |
| ☐ | ☒ | A full description of the statistical parameters including central tendency (e.g. means) or other basic estimates (e.g. regression coefficient) AND variation (e.g. standard deviation) or associated estimates of uncertainty (e.g. confidence intervals) |
| ☒ | ☐ | For null hypothesis testing, the test statistic (e.g. *F*, *t*, *r*) with confidence intervals, effect sizes, degrees of freedom and *P* value noted *Give P values as exact values whenever suitable.* |
| ☐ | ☒ | For Bayesian analysis, information on the choice of priors and Markov chain Monte Carlo settings |
| ☒ | ☐ | For hierarchical and complex designs, identification of the appropriate level for tests and full reporting of outcomes |
| ☒ | ☐ | Estimates of effect sizes (e.g. Cohen's *d*, Pearson's *r*), indicating how they were calculated |

*Our web collection on statistics for biologists contains articles on many of the points above.*

## Software and code

Policy information about availability of computer code

| Data collection | No software used for data collection |
|---|---|
| Data analysis | All code to run the analyses is available at GitHub (https://github.com/Ifeanyi-omah/Sierra_Leone_Mpox_project)<br>All software tools are listed in the Methods with version numbers: squirrel (O'Toole et al.), IQ-TREE v2.0, BEAST X v10.5.0, BEAGLE library (v3), TreeAnnotator 1.10, Tracer v1.7, LogCombiner, EpiFilter/Smooth, JUNIPER, BWA-MEM v0.7.17, SAMtools v1.6, iVar v1.3.1, Nextclade, R CODA package. |

For manuscripts utilizing custom algorithms or software that are central to the research but not yet described in published literature, software must be made available to editors and reviewers. We strongly encourage code deposition in a community repository (e.g. GitHub). See the Nature Portfolio guidelines for submitting code & software for further information.

# Data

Policy information about availability of data

All manuscripts must include a data availability statement. This statement should provide the following information, where applicable:

- Accession codes, unique identifiers, or web links for publicly available datasets
- A description of any restrictions on data availability
- For clinical datasets or third party data, please ensure that the statement adheres to our policy

All 338 MPXV genome sequences are deposited on Pathoplexus (https://doi.org/10.62599/PP_SS_232.1). Epidemiological case data are available at https://clt.npha.gov.sl/outbreak.aspx. Analysis code is available at https://github.com/Ifeanyi-omah/Sierra_Leone_Mpox_project. No other primary data were generated.

# Research involving human participants, their data, or biological material

Policy information about studies with human participants or human data. See also policy information about sex, gender (identity/presentation), and sexual orientation and race, ethnicity and racism.

| Reporting on sex and gender | Sex and age are collected via routine national surveillance. |
|---|---|
| Reporting on race, ethnicity, or other socially relevant groupings | Race and ethnicity are not routinely collected variables in Sierra Leone's public health surveillance system. Genomic data were anonymised prior to analysis. |
| Population characteristics | The study analysed n = 338 MPXV genomes from 338 individuals sampled across 14 districts in Sierra Leone, representing a geographically diverse population during the 2025 mpox outbreak.<br>Samples were obtained through routine national surveillance and outbreak response activities, encompassing both urban and rural districts. The distribution of samples reflects the national spread of mpox cases across multiple regions, as presented in the manuscript.<br>Demographic data, including age and sex, were collected as part of routine surveillance records. The affected population showed a near-equal sex distribution, with a slight skew toward younger women and older men, consistent with patterns described in the manuscript. The epidemic predominantly affected young adults (approximately 16–35 years).<br>All genomic data were anonymised prior to analysis. No sex-disaggregated genomic analyses were performed, as the study was designed as a population-level genomic investigation. |
| Recruitment | All confirmed mpox cases presenting to diagnostic facilities during the outbreak response were eligible. Samples were collected via the national surveillance system. Self-selection bias: cases from districts with better laboratory infrastructure (Western Area Urban, Kenema, Port Loko) are overrepresented; mild/asymptomatic cases not presenting to healthcare facilities are likely underrepresented.<br>All samples were collected between 10th January and 3rd August 2025 from outbreak response surveillance. The dataset was produced through collaborative efforts between the Central Public Health Reference Laboratory (CPHRL), Kenema Government Hospital (KGH) Viral Hemorrhagic Fever (VHF) Laboratory, and the Institut Pasteur de Dakar (IPD) mobile laboratory in Port-Loko. Samples were processed for DNA extraction at CPHRL and KGH VHF lab |
| Ethics oversight | This work was conducted as part of the public health response to the mpox outbreak under the mandate of the National Public Health Agency (NPHA), Ministry of Health, Sierra Leone, and all activities were performed in accordance with national outbreak investigation guidelines. As this investigation formed part of an emergency public health response, it was exempt from institutional ethics review and the requirement for individual informed consent under national public health regulations. All data were anonymized prior to analysis to protect patient confidentiality. All authors meet established authorship criteria and contributed substantially to study design, data generation, analysis, interpretation, and manuscript preparation. Local public health scientists and laboratory personnel were actively involved throughout the investigation and analysis, ensuring equitable collaboration, local ownership, and appropriate recognition of contributions. Care was taken to minimize stigma and harm in data handling, analysis, and reporting. |

Note that full information on the approval of the study protocol must also be provided in the manuscript.

# Field-specific reporting

Please select the one below that is the best fit for your research. If you are not sure, read the appropriate sections before making your selection.

☒ Life sciences    ☐ Behavioural & social sciences    ☐ Ecological, evolutionary & environmental sciences

For a reference copy of the document with all sections, see nature.com/documents/nr-reporting-summary-flat.pdf

# Life sciences study design

All studies must disclose on these points even when the disclosure is negative.

| | |
|---|---|
| Sample size | We generated 338 high-quality MPXV genomes from cases identified across the country between January 10 and August 3, 2025 (Figure 1A). The dataset was produced through collaboration among the Central Public Health Reference Laboratory (CPHRL, n=128 sequences), the Kenema Government Hospital (KGH) Viral Hemorrhagic Fever (VHF) Laboratory (n=105), and the Institut Pasteur de Dakar (IPD) mobile laboratory in Port-Loko (n=105). We also included two genomes from the Western Area Urban district sequenced by the Beijing Institute of Technology, bringing the total to 340 genomes<br>No formal power calculation was performed; this is a total-population genomic surveillance study where the sample size is determined by the epidemic itself, consistent with standard practice in genomic epidemiology. |
| Data exclusions | PCR-confirmed mpox cases with Ct <=30 and genome coverage >60% during the outbreak period. All qualifying samples from 10 Jan to 3 Aug 2025 were included (n=338) |
| Replication | Three chains of each Bayesian phylogenetic reconstruction was performed to assess whether they converge on the same posterior. They did. Uncertainty is quantified across the three combined independent chains, and across 10 000 phylogenetic trees in the posterior.This is an observational genomic epidemiology study with no experimental replication in the traditional sense. Phylogenetic and phylodynamic analyses were run as duplicate independent MCMC chains; both chains converged (ESS >200) and results were consistent. JUNIPER was run across two chains with consistent posterior distributions |
| Randomization | Irrelevant as there are no experimental groups. |
| Blinding | Irrelevant as metadata is vital to analyses. |

# Reporting for specific materials, systems and methods

We require information from authors about some types of materials, experimental systems and methods used in many studies. Here, indicate whether each material, system or method listed is relevant to your study. If you are not sure if a list item applies to your research, read the appropriate section before selecting a response.

### Materials & experimental systems

| n/a | Involved in the study |
|---|---|
| ☒ ☐ | Antibodies |
| ☒ ☐ | Eukaryotic cell lines |
| ☒ ☐ | Palaeontology and archaeology |
| ☒ ☐ | Animals and other organisms |
| ☒ ☐ | Clinical data |
| ☒ ☐ | Dual use research of concern |
| ☒ ☐ | Plants |

### Methods

| n/a | Involved in the study |
|---|---|
| ☒ ☐ | ChIP-seq |
| ☒ ☐ | Flow cytometry |
| ☒ ☐ | MRI-based neuroimaging |

## Plants

| | |
|---|---|
| Seed stocks | *Report on the source of all seed stocks or other plant material used. If applicable, state the seed stock centre and catalogue number. If plant specimens were collected from the field, describe the collection location, date and sampling procedures.* |
| Novel plant genotypes | *Describe the methods by which all novel plant genotypes were produced. This includes those generated by transgenic approaches, gene editing, chemical/radiation-based mutagenesis and hybridization. For transgenic lines, describe the transformation method, the number of independent lines analyzed and the generation upon which experiments were performed. For gene-edited lines, describe the editor used, the endogenous sequence targeted for editing, the targeting guide RNA sequence (if applicable) and how the editor was applied.* |
| Authentication | *Describe any authentication procedures for each seed stock used or novel genotype generated. Describe any experiments used to assess the effect of a mutation and, where applicable, how potential secondary effects (e.g. second site T-DNA insertions, mosiacism, off-target gene editing) were examined.* |

