## [Peer Review File · Nature Medicine]

Genomic epidemiology of the 2025 mpox epidemic in Sierra Leone

Corresponding Author: Mr Allan Campbell

Version 0:

Reviewer comments:

Reviewer #1

(Remarks to the Author)

Campbell et al. present a highly relevant and timely study that sheds important light on mpox epidemics in West Africa. The manuscript is rich in detail, technically strong, and well supported by the literature. It provides clear phylogenomic insights and delivers them in a well-structured manner consistent with current work in pathogen genomics and outbreak analytics. The writing is accessible, and the scientific reasoning is sound and compelling. The conclusions are appropriate, a part one sentence in the abstract (see below). Congratulations to the authors for this collaboration across multiple laboratories to provide a great overview of the genomic epidemiology of mpox in Sierra Leone.

Specific Comments

- Please consider adding line or section numbers in the next submission to facilitate referencing.
- Some affiliations appear incomplete and should be reviewed.

Abstract

The abstract should be revised to better highlight the main findings and ensure that the discussion directly connects to those results. At present, it concludes that the findings “underscore that effective control remains achievable through targeted vaccination, strengthened early warning systems, and improved access to genomic and diagnostic surveillance.” However, the results do not demonstrate that targeted vaccination—or any specific control measure or event (eg depletion of susceptible population?)—drove the rapid decline in cases to zero. These statements may be appropriate for the background or discussion, but they are not supported by the data presented.

Background

- Several paragraphs are long (12–15 lines) and combine epidemiology, phylogenetics, and historical context. Breaking these into smaller, thematic units would improve readability.
- In the global outbreak, >90% of cases occurred among MSM; please consider mentioning this with a citation.
- Clade I is now subdivided into clades Ia and Ib and should be noted.
- It remains unclear why Sierra Leone experienced such rapid and explosive growth compared to neighbouring countries (e.g., Ghana, Côte d’Ivoire). Adding a brief explanation—mobility patterns, early undetected spread, or other contributing factors—would strengthen this section. This is especially important if the authors argue that intrinsic viral differences are unlikely to generate large differences in outbreak dynamics.
- When discussing limitations in genomic data, including a concrete example—missing early samples, limited temporal depth, or uneven district-level sampling—would help clarify the issue.
- The point that demographic patterns may reflect network structure rather than intrinsic viral characteristics is important. However, historical and recent epidemiological evidence (e.g., PALM007 mortality data) and laboratory findings do indicate differences in severity between clades Ia, Ib and IIb (we know little about clade IIa associated severity). Because of this, caution is warranted to avoid oversimplification.

Results

- Some portions of the Results appear in bold text and may need formatting correction.
- The discussion of the Clade IIa outlier is interesting but disproportionately long. Consider shortening or relocating it to the Supplementary Materials.
- The manuscript cites an introduction rate of ~6 per year. If possible, please provide the credible interval. It would also be helpful to clarify whether this rate is consistent across clades or lineages capable of sustained human-to-human transmission.
- The section on the long stem is handled well, but adding a sentence outlining alternative scenarios—such as unsampled emergence in a neighbouring country followed by rapid introduction into Freetown—would further contextualize the

uncertainty.

- A brief explanation of how “transmission chains” are defined—particularly for readers unfamiliar with Markov jump reconstructions—would enhance accessibility.
- The negative correlation between persistence and origin time is intriguing. A concise interpretive statement (e.g., later chains being shorter due to stronger interventions or diminishing susceptible populations) would strengthen the narrative.
- Please ensure that map references appear at the point where the corresponding geographic interpretation is discussed to maintain narrative flow.

Discussion

- Is the G.1 outbreak unique to Sierra Leone, or were there similar dynamics in neighbouring countries such as Côte d’Ivoire or Ghana? (See earlier comments in the Background section.)
- The statement “surveillance systems detected approximately 49% of total cases” might be better phrased as “approximately half” to reflect uncertainty around the exact estimate.
- It is unclear whether the authors infer undetected spread within Sierra Leone before detection or whether the missing intermediate lineage may have circulated elsewhere. Based on contextual knowledge, the authors could expand on diagnostic preparedness, response timing, and whether few months of silent circulation in Sierra Leone is plausible.
- The scarcity of genomic data from other West African countries should be explicitly noted if it limits interpretation.
- Please specify which groups were prioritized for vaccination and clarify which populations were considered high-risk.
- Briefly introduce the enhanced integrated mpox response to give readers context.
- Although the paper is not focused primarily on Sierra Leone’s outbreak dynamics, a deeper discussion of the outbreak’s explosive nature—and the factors that enabled control down to zero weekly cases—would significantly strengthen the discussion. The speed and completeness of the decline was exceptional and differs notably from the 2022 outbreaks in Europe and the Americas, where elimination has been challenging and low-level circulation persisted.
- Discuss why such a rapid and sustained return to zero occurred, and how this contrasts with patterns observed elsewhere. It would be interesting to read authors’ view on potential depletion of susceptible population

Methods

- Please avoid reintroducing acronyms multiple times.

(Remarks on code availability)

Reviewer #2

(Remarks to the Author)

This manuscript reports an investigation in the emergence and spread of clade II mpox virus in Sierra Leone in 2025. The authors sequenced hundreds of viral genomes from this epidemic and analyze them with State of the Art phylodynamic and phylogeographic inference tools. The findings are mostly clear, though I detail below areas where the authors could extend their analyses to provide more robustness for their conclusions.

Regarding APOBEC SNPs depicted in Figure 1, not all of these substitutions are necessarily the result of APOBEC-mediated hypermutation. One quarter of all regular C to T mutations will occur adjacent to another T (CT to TT). Rather, these SNPs are consistent with APOBEC activity. APOBEC-like mutations will be enriched when replicating in humans, but they would be expected in rodent populations. It would be helpful to emphasize it is the enrichment, not the presence that supports sustained human transmission.

Further, I think the manuscript would benefit from a formal assessment of APOBEC enrichment along lineages to determine when and where these signatures provide evidence of sustained human transmission. It is currently too descriptive.

What kind of robustness checks can be applied to the inference that G.1 first became established in the Western Area Urban region. This region is the most represented in the phylogeny, and it is not clear to me that the analysis presented adequately addresses the authors’ stated concern about “under-ascertainment of transmission elsewhere during the early phase of the outbreak”.

I have similar concerns for the persistence analysis. How robust are these conclusions to biases in diagnosis rates and subsequent genome sequencing rates?

The Results section lacks any indication of vaccine success/impact, the description of which the Discussion post-dates the genome sampling. I am left unclear on how to evaluate any statements regarding the impact of vaccines. Though I am wont to agree with the sentiment expressed in the Discussion, this paper does not provide evidence in favor of these conclusions.

Minor comments:

Figure 1A. I don’t see an annotation about the start and/or location of vaccination campaigns. And the second y-axis is not labeled.

The description of Supplementary Figure 1 references a basal node corresponding with a Rotterdam sequence and a clade lying downstream of the 198 sequence. These descriptions, focused on sampled viruses rather than phylogenetic relationships are imprecise and inaccurate. I cannot reconcile this description with the tree depicted.

The inferred epidemic size should not be described in terms of cases. A case is a diagnosed and reported infection in an individual. An undiagnosed case is a contradiction in terms.

(Remarks on code availability)

Version 2:

Reviewer comments:

Reviewer #1

(Remarks to the Author)

The author appropriately addressed all comments. I have a problem with the following statement:

"However, there is a difference among these Clades in terms of clinical severity. Clade I, for example, has historically been associated with higher case fatality ratios (CFRs) than clade II (Cho et al. 2024; Beer and Rao 2019), (Beer and Rao 2019; Wawina-Bokalanga et al. 2025). Recent trial data from the DRC highlight differential patient outcomes with the recommended tecovirimat drugs, unless patients are hospitalised and receive highquality supportive care. (Postal et al. 2025; PALM007 Writing Group et al. 2025) The clinical severity of Clade IIa infection in humans is still less well characterised than that of Clades Ia/Ib and IIb."

Although Clade I has historically been associated with more severe disease, more recent data indicate important heterogeneity within this lineage. Accumulating epidemiological evidence does not demonstrate remarkable mortality differences between Clade Ib and Clade IIb. In the PALM tecovirimat trial, Clade Ia mortality was 1.7%, markedly lower than the ~10% CFR frequently cited in earlier literature. Reported mortality for Clade Ib has generally remained below 1% in most affected countries.

While mortality may be higher for Clade Ia compared with other subclades, there remains considerable uncertainty as to whether this reflects intrinsic viral virulence or differences in the affected populations. Clade Ia transmission has predominantly occurred among children in endemic regions, where contextual factors—such as malnutrition, co-infections, delayed healthcare access, and limited supportive care—may substantially influence observed outcomes.

Experimental laboratory data do suggest greater virulence of Clade I viruses relative to Clade II. However, contemporary epidemiological data do not consistently demonstrate large differences in clinical severity across subclades, particularly when healthcare access and supportive management are accounted for.

For accuracy and precision, it would therefore be preferable to modify the sentence to explicitly refer to Clade Ia rather than Clade I as a whole, reflecting the comparatively low reported mortality associated with Clade Ib.

(Remarks on code availability)

Reviewer #2

(Remarks to the Author)

The authors have substantially improved their manuscript and adequately responded to the criticisms raised by myself and the other reviewer. I have no further suggestions.

(Remarks on code availability)

Response to Reviewers for Campbell et al:

Reviewer #1 (Remarks to the Author):

Campbell et al. present a highly relevant and timely study that sheds important light on mpox epidemics in West Africa. The manuscript is rich in detail, technically strong, and well supported by the literature. It provides clear phylogenomic insights and delivers them in a well-structured manner consistent with current work in pathogen genomics and outbreak analytics. The writing is accessible, and the scientific reasoning is sound and compelling. The conclusions are appropriate, a part one sentence in the abstract (see below). Congratulations to the authors for this collaboration across multiple laboratories to provide a great overview of the genomic epidemiology of mpox in Sierra Leone.

We appreciate the constructive and insightful comments, which have helped us clarify and strengthen the manuscript.

Specific Comments:

Please consider adding line or section numbers in the next submission to facilitate referencing.

We thank the reviewer for bringing this oversight to our attention. We have now added line numbers.

Some affiliations appear incomplete and should be reviewed.

We have reviewed all affiliations and assigned them appropriately.

Abstract

The abstract should be revised to better highlight the main findings and ensure that the discussion directly connects to those results. At present, it concludes that the findings “underscore that effective control remains achievable through targeted vaccination, strengthened early warning systems, and improved access to genomic and diagnostic surveillance.” However, the results do not demonstrate that targeted vaccination—or any specific control measure or event (eg depletion of susceptible population?)—drove the rapid decline in cases to zero. These statements may be appropriate for the background or discussion, but they are not supported by the data presented.

We thank the reviewers for their comments. We acknowledge that we did not demonstrate the impact of vaccination with data directly, and therefore, we have rephrased the abstract to focus more on the data and analysis we presented in this work (lines 72-92). The revised abstract now reads:

“In January 2025, Sierra Leone reported its first mpox case in eight years, followed by a rapid nationwide surge that made it the epicentre of the continental mpox virus (MPXV) Clade IIb A.2.2 outbreak, with more than 5,000 confirmed cases reported by August. To investigate the origin, timing, and spread of this epidemic, we generated 338 high-quality MPXV genomes from cases across 14 districts and conducted Bayesian phylogeographic analyses. We demonstrate that the outbreak was driven by a newly emerging Clade IIb/sh lineage, G.1 (A.2.2.1), which descended from lineages circulating in the ongoing Nigerian epidemic and exhibited strong APOBEC3 mutational enrichment, consistent with sustained human-to-human transmission. We estimated the probable time of emergence of G.1 tMRCA to 27 September 2024 (95% HPD: 12 August–11 November 2024), indicating ~3 months of cryptic circulation before detection in early January 2025 and rapid early epidemic growth (doubling time ~3 weeks). Phylogeographic reconstructions identify Western Area Urban as the dominant hub of persistence and dissemination, generating repeated introductions that seeded outbreaks across the country and multiple international export events. Phylodynamic inference further suggests substantial under-ascertainment, with an estimated total epidemic size of ~10,400 cases compared with 5,328 laboratory-confirmed cases during the sampling period. Together, these findings reveal the origins and dispersal dynamics of the 2025 mpox outbreak in Sierra Leone, underscoring the need for strengthened early warning systems and improved access to genomic and diagnostic surveillance to prevent epidemics.”

Background

Several paragraphs are long (12–15 lines) and combine epidemiology, phylogenetics, and historical context. Breaking these into smaller, thematic units would improve readability.

We have restructured the Background section into smaller, thematic paragraphs as suggested by the reviewer. The text now progresses logically from global mpox classification and epidemiological shifts to the specific context of Sierra Leone, potential drivers of the explosive spread, genomic and demographic characterisation, and the study's objectives.

In the global outbreak, >90% of cases occurred among MSM; please consider mentioning this with a citation.

We have added this to the fifth paragraph, explicitly noting that over 90% of cases in the 2022 global outbreak occurred among men who have sex with men (MSM), with appropriate citations included. Lines 133-134 now read: *"Unlike the early 2022 Clade IIb/sh2017 global wave, with over 90% of cases reported among men who have sex with men (MSM) community (Paredes et al. 2024; Gigante et al. 2022; Iñigo Martínez et al. 2022)"*

Clade I is now subdivided into clades Ia and Ib, and should be noted.

We have updated the first paragraph to explicitly reflect the subdivision of Clade I into subclades Ia and Ib, as well as the subdivision of Clade II into IIa and IIb. Revised lines 100-103 now read: *"MPXV is classified into two main clades: I and II (Ruis et al. 2025; Happi et al. 2022). Historically, Clade I, now subdivided into Ia and Ib, was confined to zoonotic infections arising from Central African animal reservoirs, whereas Clade II (also subdivided into IIa and IIb) was restricted to West Africa and associated with lower case fatality rates (Okwor et al. 2023)."*

It remains unclear why Sierra Leone experienced such rapid and explosive growth compared to neighbouring countries (e.g., Ghana, Côte d'Ivoire). Adding a brief explanation... would strengthen this section.

We have added a paragraph addressing this disparity in both the Background. We suggest that the explosive growth in Sierra Leone, relative to neighbouring countries, is consistent with amplification within the dense and highly mobile population of Western Area Urban. We further note that the early undetected spread within these high-mobility networks may have enabled the virus to seed multiple regions before surveillance

systems were fully activated, outpacing what contact tracing could manage. Lines 121-128 now read:

“The rapid and explosive growth of the outbreak in Sierra Leone, compared to neighbouring countries such as Ghana, Liberia, Guinea or Côte d’Ivoire, likely stems from specific socio-demographic drivers. Freetown’s dense and highly mobile population (~1.4 million of the nation’s 8.6 million residents) facilitated the inter-district spread, which outpaced the deployment of interventions such as testing, case isolation, and contact tracing. Furthermore, early undetected spread within these high-mobility networks may have allowed the virus to seed multiple regions before surveillance systems were fully activated.”

When discussing limitations in genomic data, including a concrete example—missing early samples, limited temporal depth, or uneven district-level sampling—would help clarify the issue.

We have expanded the limitations section to include these examples. We now explicitly cite missing samples from the early phases of the outbreak, uneven district-level sampling, and limited temporal depth as key challenges that complicate the inference of exact origins. Lines 151-153 In the revised text now read: *“Notably, specific limitations, such as the absence of samples from the early phases of the outbreak, uneven district-level sampling, and limited temporal sampling range, make it challenging to more precisely resolve the origins of the G.1 outbreak. Consequently, while our data point to local transmission, we cannot exclude the possibility that G.1 emerged elsewhere before being introduced to Sierra Leone. It therefore remains uncertain exactly when and where the emergence occurred, as well as the mechanisms underlying its rapid nationwide spread.*

The point that demographic patterns may reflect network structure rather than intrinsic viral characteristics is important. However... caution is warranted to avoid oversimplification.

We appreciate this important nuance. We have revised the text to avoid oversimplification. While we discuss the role of contact networks in shaping demographic profiles, we now also explicitly recognise intrinsic differences in clinical severity, citing higher case fatality ratios (CFRs) in Clade I and recent trial data (e.g., PALM007) that indicate differential virulence. Revised lines 143-148 now read:

“However, there is a difference among these Clades in terms of clinical severity. Clade I, for example, has historically been associated with higher case fatality ratios (CFRs) than clade II (Cho et al. 2024; Beer and Rao 2019), (Beer and Rao 2019; Wawina-Bokalanga et al. 2025). Recent trial data from the DRC highlight differential patient outcomes with the recommended tecovirimat drugs, unless patients are hospitalised and receive high-quality supportive care. (Postal et al. 2025; PALM007 Writing Group et al. 2025) The clinical severity of Clade IIa infection in humans is still less well characterised than that of Clades Ia/Ib and IIb.”

Results

Some portions of the Results appear in bold text and may need formatting correction.

We thank the reviewer for this observation. The text has been reformatted to ensure consistent styling throughout the entire manuscript.

The discussion of the Clade IIa outlier is interesting but disproportionately long. Consider shortening or relocating it to the Supplementary Materials.

We agree with the reviewer and have relocated the detailed results (lines 941-962) of the Clade IIa sequence to the Supplementary Materials.

The manuscript cites an introduction rate of ~6 per year. If possible, please provide the credible interval. It would also be helpful to clarify whether this rate is consistent across clades or lineages capable of sustained human-to-human transmission.

We have updated the text to include the credible interval of the estimated rate of APOBEC3 accumulation. We also clarified that this rate is consistent across MPXV lineages associated with sustained human-to-human transmission, including Clade IIb/sh2017 Lineage A and its descendants (O'Toole et al. 2023), as well as the recently described Clade Ib/sh2023 (Wawina-Bokalanga et al. 2025). Revised lines 231-233 now read: *“Previous studies have estimated that APOBEC3-like mutations accumulate at a rate of approximately 6 per year [95% credible interval 5-7], a rate consistent across MPXV clades and lineages with sustained human transmission (Parker et al. 2025; O'Toole et al. 2023; Wawina-Bokalanga et al. 2025).”*

The section on the long stem is handled well, but adding a sentence outlining alternative scenarios—such as unsampled emergence in a neighbouring country followed by rapid introduction into Freetown—would further contextualise the uncertainty.

We have added a sentence explicitly noting that the long branch likely reflects unsampled circulation in Nigeria or a neighbouring country, with a subsequent period of approximately three months of undetected transmission following introduction into Sierra Leone. Revised lines 332-334 now read: *“The long stem branch of G.1 suggests an extended period of undetected circulation. This may reflect cryptic circulation in a neighbouring country followed by an introduction, or unsampled transmission within Sierra Leone itself. Regardless of the precise geographic source, the data implies potentially ~3 months of undetected local circulation before identification.”*

A brief explanation of how “transmission chains” are defined—particularly for readers unfamiliar with Markov jump reconstructions—would enhance accessibility.

We have added a clear definition to the Results Section to aid accessibility. We now define transmission chains in this context as “continuous periods of viral lineage circulation within a specific location following an introduction event,” derived from the statistical reconstruction of location changes along phylogenetic branches (Markov jumps). Revised lines 342-346. Now read: *“To investigate this pattern of within-country spatiotemporal spread while accounting for uncertainty in phylogeographic reconstruction due to sparse sampling, we incorporated a Markov jump counting approach. This method reconstructs the history of location changes along the phylogenetic branches, allowing us to estimate the timing and origin of geographic transmission chains (Lemey et al. 2009; Lemey et al. 2014), defined here as continuous periods of viral lineage circulation within a specific location following an introduction event both at the regional and district levels.”*

The negative correlation between persistence and origin time is intriguing. A concise interpretive statement (e.g., later chains being shorter due to stronger interventions or diminishing susceptible populations) would strengthen the narrative.

We have strengthened the narrative by explicitly linking the shorter transmission chains observed later in the outbreak to the timing of public health interventions. We note that the shift toward shorter chains coincides with the rollout of vaccination campaigns and the strategic shift to mandatory treatment by isolation, consistent with the fragmentation

of transmission networks. We also provided a caveat about our sampling artefact. Revised lines 424-435 now read:

“The strong negative correlation observed between the duration of transmission chains and the epidemic phase (Figure 4B) indicates that these transmission chains became shorter as the outbreak progressed. This pattern is consistent with the timeline of public health interventions. This notably includes the vaccination campaigns that began in the Western Area Urban and Rural districts on March 26 and expanded nationwide by April 30, alongside the ring vaccination of contacts initiated in early June (Figure 1A). While these campaigns reduced the susceptible population, an integrated phase was initiated after it was observed that home-based management was unintentionally sustaining viral transmission. The subsequent transition to mandatory isolation may have played a crucial role in breaking transmission networks and preventing community clusters (Figure 4B).”

We have revised Figure 1A to include intervention timelines.

Revised Figure 1: A) Confirmed mpox incidence in Sierra Leone during 2025, showing the sequencing rate of this study over time (bars) and the time-varying reproductive number (line, right y-axis, with 95% confidence intervals). Annotations mark the start of vaccination campaigns in Western Area Urban (W.A.U.) and Western Area Rural (W.A.R.), as well as the launch of the national vaccination campaign.

Please ensure that map references appear at the point where the corresponding geographic interpretation is discussed to maintain narrative flow.

We have corrected the figure citations to ensure that map references (Figures 4C) appear immediately alongside the relevant geographic interpretations in the text.

Discussion

Is the G.1 outbreak unique to Sierra Leone, or were there similar dynamics in neighbouring countries such as Côte d'Ivoire or Ghana? (See earlier comments in the Background section.)

We thank the reviewer for this important question regarding regional dynamics. During the period when the Sierra Leone outbreak was rapidly expanding, there were no reports of a concurrent large-scale surge in mpox cases in neighbouring countries, such as Ghana, Liberia, Guinea, or Côte d'Ivoire (Mathieu et al. 2022), although the dynamics of the transmission are still poorly characterised in these countries. Prior to this surge and during our study window, the G.1 lineage was primarily detected in Sierra Leone, and international cases identified elsewhere were epidemiologically linked to travel from Sierra Leone.

Comparison of the MPXV case reports from both prior to and during G.1 outbreaks in a neighbouring country to Sierra Leone, illustrating the disproportionate number of mpox cases in Sierra Leone compared to others, adapted from Mathieu et al. (2022).

We have expanded the text to discuss possible contributing factors to the explosive patterns in Sierra Leone. (i) a period of cryptic transmission prior to the first confirmed case; (ii) disparities in regional testing access and reporting; and (iii) amplification within the dense, highly mobile population of the capital city (Western Area Urban). While other neighbouring countries' capitals share similar demographic densities, the escalation of an outbreak is inherently stochastic: high density provides the necessary conditions for rapid spread, but an epidemic requires a successful introduction into high-risk transmission networks. Although neighbouring countries reported relatively few cases before this surge in Sierra Leone, the lack of comparable genomic data limits our ability to assess whether they experienced similar dynamics or why this transmission is unique to Sierra Leone. However, available data from Nextstrain indicate that different lineages were circulating nearby: Ghana predominantly reported lineage A.2.5 (likely imported from Nigeria), while Liberia reported Clade Ila (Nyan et al. 2025). The lineages in Côte d'Ivoire remain poorly characterised. While we cannot definitively explain why outbreaks in these neighbouring high-density capitals did not escalate to the same degree, phylogenetic evidence indicates that the virus likely circulated undetected in the region before finding the specific foothold required for explosive establishment in Freetown. Revised lines 469-482 now read:

“Our phylogeographic analyses indicate that G.1 was most likely established in the Western Area Urban district, which served as the primary and persistent source for viral dissemination across Sierra Leone. While neighbouring capitals share similar densities and more cases of mpox have been reported in those countries in late 2025 (Mathieu et al. 2022), we suggest that this specific escalation in Freetown reflects the stochastic nature of disease emergence. Density provided the necessary conditions for amplification, but the specific introduction into high-risk networks was required to trigger an outbreak of this scale, which may be the reason for the limited scale in nearby countries. This hypothesis is corroborated by epidemiological data that showed that the early cases were concentrated within sexual transmission networks, subsequent spread became more generalised, maintained through both household and community contact transmission alongside sexual networks—a pattern paralleling the recent Clade Ib expansion in the DRC, Burundi, and Uganda (Kangbai et al. 2025; Mitjà et al. 2025). At the time of writing, only Guinea has reported imported cases of G.1 from the ongoing outbreak in Sierra Leone. Meanwhile, neighbouring countries reported limited mpox activity during the early phase of G.1 expansion, with distinctly different lineages: Liberia

reported Clade IIa cases (Nyan et al. 2025), and Ghana primarily reported Clade IIa and Clade IIb lineage A.2.5 (Agbodzi et al. 2026), indicative of imports from ongoing low-level transmission in Nigeria (Parker et al. 2025). Consequently, the extent of G.1's spread in neighbouring countries across West Africa remains largely uncharacterised, and it is hoped that future sequencing will clarify this issue. "

District-level incidence and genomic sampling in Sierra Leone (Jan–Aug 2025). A) For each district, daily mpox cases (7-day rolling mean; line) and weekly counts of genomes generated (bars, inverted for visual separation) are shown on a shared time axis. Traces are coloured by district using a consistent palette; y-axes are comparable within rows, with labels shown only for the first column. B) Administrative map of Sierra Leone at the district level, colored and labelled to match panel A, providing geographic context for the time-series panels.

The statement “surveillance systems detected approximately 49% of total cases” might be better phrased as “approximately half” to reflect uncertainty around the exact estimate.

We agree with the reviewer that this phrasing better reflects the inherent uncertainty in these estimates. We have revised the manuscript to read "approximately half" instead of "approximately 49%" (lines 494).

It is unclear whether the authors infer undetected spread within Sierra Leone before detection or whether the missing intermediate lineage may have circulated elsewhere. Based on contextual knowledge, the authors could expand on diagnostic preparedness, response timing, and whether a few months of silent circulation in Sierra Leone is plausible.

We have expanded the discussion to note that the ~3 months of undetected circulation within Sierra Leone likely allowed the virus to establish multiple transmission chains before the first confirmed case was identified. This surveillance gap is consistent with the rapid initial case growth that outpaced early contact tracing efforts, underscoring the critical need for improved diagnostic preparedness and earlier detection capabilities (lines 494-502).

“We estimate that lineage G.1 cryptically circulated locally for approximately three months before detection, spreading to at least three districts beyond the Western Area Urban district during this period. However, the long stem branch connecting G.1 to its closest relative (lineage A.2.2.1 from Togo) reveals a significantly deeper history: the estimated tMRCA dates to more than a year before the lineage’s establishment in Sierra Leone around September 2024. This temporal gap is consistent with prolonged circulation in an unsampled location within West Africa prior to its introduction. Consequently, while the data suggest a regional origin, current genomic undersampling prevents us from inferring the specific country of direct introduction.”

The scarcity of genomic data from other West African countries should be explicitly noted if it limits interpretation.

We have explicitly noted this limitation in the revised Discussion. Specifically, we highlight the temporal gap between the estimated tMRCA of the G.1 lineage and its closest relative (A.2.2.1 from Togo); August 9, 2023 [95% HPD: April 3, 2023 – December 19, 2023], and the establishment of G.1 in Sierra Leone around September 2024. This gap of more than one year is consistent with circulation in an unsampled location within West Africa prior to introduction into Sierra Leone. We emphasise that current undersampling in the region precludes inference of the specific country of direct introduction; intensifying surveillance in neighbouring countries may help resolve this uncertainty (lines 494-502).

“We estimate that lineage G.1 cryptically circulated locally for approximately three months before detection, spreading to at least three districts beyond the Western Area Urban district during this period. However, the long stem branch connecting G.1 to its closest relative (lineage A.2.2.1 from Togo) reveals a significantly deeper history: the estimated tMRCA dates to more than a year before the lineage’s establishment in Sierra Leone around September 2024. This temporal gap is consistent with prolonged circulation in an unsampled location within West Africa prior to its introduction. Consequently, while the data suggest a regional origin, current genomic undersampling prevents us from inferring the specific country of direct introduction.”

Please specify which groups were prioritised for vaccination and clarify which populations were considered high-risk.

We have added specific details regarding vaccination prioritisation to the manuscript. The groups defined as high-risk and prioritised for vaccination included: sex workers, men who have sex with men (MSM), contacts of confirmed mpox cases (particularly sexual partners and household members), people living with HIV, frontline response teams, individuals reporting multiple sexual partners, and healthcare workers. Revised lines 523-525 now read:

“Vaccination coverage included health-care workers (23%), identified contacts (43%), high-risk groups (12%), defined as sex workers, men who have sex with men (MSM), contacts and households of confirmed cases, people living with HIV, and individuals with multiple sexual partners and others (22%).”

Briefly introduce the enhanced integrated mpox response to give readers context.

We have introduced the Enhanced Integrated Mpox Response (EIMR) in the text to provide necessary context. This response involved the national expansion of vaccination to the risk groups listed above, extending beyond the initial focus on Western Area Urban/Rural, as well as increased genomic surveillance. We also clarify that a key component of the enhanced response was a strategic shift in case management. The Ministry of Health identified that home-based isolation may have inadvertently sustained transmission. Consequently, the strategy shifted to mandatory treatment by isolation, whereby active cases were removed from the community to treatment centres. Together with the vaccination campaigns (initiated on March 26 in Western Area Urban and expanded nationwide by April 30), these measures coincided with a marked reduction in

transmission and fragmentation of transmission networks. Revised lines 531-541 now read:

“This enhanced phase was initiated by the Ministry of Health, which observed that home-based management was inadvertently sustaining transmission networks despite decreasing the number of susceptible individuals among high-risk groups. The subsequent strategic shift to mandatory treatment by isolation is credited by national authorities as a primary driver of the rapid decline in incidence. This distinct approach likely facilitated the rapid return to zero cases, contrasting with the lingering low-level circulation often observed in Nigeria among the human population (Parker et al., 2025) and the DRC, where zoonotic transmission continues to reseed new human infections (Wawina-Bokalanga et al., 2025). Together, these actions underscore the importance of layered, timely vaccination strategies, effective case isolation supported by robust surveillance, in controlling transmission.”

Although the paper is not focused primarily on Sierra Leone’s outbreak dynamics, a deeper discussion of the outbreak’s explosive nature—and the factors that enabled control down to zero weekly cases—would significantly strengthen the discussion. Discuss why such a rapid and sustained return to zero occurred, and how this contrasts with patterns observed elsewhere.

We have significantly strengthened the Discussion section to address the "explosive" nature of the outbreak and the specific factors associated with its rapid return to zero. We describe how the proactive and coordinated public health response in Sierra Leone contrasted with patterns observed in settings such as DRC and Nigeria, where ongoing zoonotic spillover has sustained low-level human transmission.

In Sierra Leone, we highlight the strategic shift to mandatory treatment by isolation as a key distinguishing feature of the response. By removing infectious individuals from their households and contact networks, this approach is consistent with the rapid fragmentation of transmission chains. We conclude that this combination of widespread vaccination, ring vaccination of contacts, and mandatory isolation depleted the susceptible population and fragmented transmission networks more effectively than less stringent measures used elsewhere. Revised Lines 531-541 now read:

“This enhanced phase was initiated by the Ministry of Health, which observed that home-based management was inadvertently sustaining transmission networks despite decreasing the number of susceptible individuals among high-risk groups. The subsequent strategic shift to mandatory treatment by isolation is credited by national authorities as a primary driver of the rapid decline in incidence. This distinct approach likely facilitated the rapid return to zero cases, contrasting with the lingering low-level circulation often observed in different parts of Nigeria among the human population (Parker et al., 2025) and the DRC, where zoonotic transmission continues to reseed new human infections (Wawina-Bokalanga et al., 2025). Together, these actions underscore the importance of layered, timely vaccination strategies and effective case isolation, supported by robust surveillance, in controlling the disease's transmission.

Methods

Please avoid reintroducing acronyms multiple times.

We thank the reviewer for this keen observation. We have revised the Methods section to remove all redundant acronyms.

Reviewer #2 (Remarks to the Author):

This manuscript reports an investigation into the emergence and spread of clade II mpox virus in Sierra Leone in 2025. The authors sequenced hundreds of viral genomes from this epidemic and analyzed them with State of the Art phylodynamic and phylogeographic inference tools. The findings are mostly clear, though I detail below areas where the authors could extend their analyses to provide more robustness for their conclusions.

We thank the reviewer for their careful review and insightful comments.

Regarding APOBEC SNPs depicted in Figure 1, not all of these substitutions are necessarily the result of APOBEC-mediated hypermutation. One quarter of all regular C to T mutations will occur adjacent to another T (CT to TT). Rather, these SNPs are consistent with APOBEC activity. APOBEC-like mutations will be enriched when replicating in humans, but they would be expected in rodent populations. It would be helpful to emphasize it is the enrichment, not the presence that supports sustained human transmission.

We thank the reviewer for this important observation. We agree that it is the enrichment of APOBEC-context mutations, rather than their mere presence, that supports sustained human-to-human transmission. We have revised the text to clarify this distinction and to place APOBEC signatures in their appropriate evidentiary context.

As described previously (O'Toole et al., 2023; Parker et al., 2025), although the precise timing of the zoonotic jump cannot be determined, multiple lines of evidence suggest

that this lineage has been maintained through human-to-human transmission since its emergence. These include epidemiological data, contact tracing, and the documented international spread of the ancestral lineage (A.2.1) via human travel.

In this context, APOBEC3 signatures provide supporting evidence. In the natural reservoir rodent hosts, where APOBEC3 is absent, all mutations arise from polymerase error, with only a small fraction (~7%) occurring in TC→TT or GA→AA contexts. In contrast, in human hosts, APOBEC3-driven mutations are strongly enriched, occurring at rates estimated to be ~28-fold higher than those due to polymerase error alone. Because only a small subset of polymerase errors occur in APOBEC3 target contexts, enrichment of these motifs is highly informative when interpreted alongside epidemiological evidence.

We have revised the manuscript to make this reasoning explicit in lines 192-201

“The enrichment of APOBEC3-context mutations, rather than their mere presence, molecular marker that distinguishes sustained human transmission from zoonotic spillover events (O’Toole et al. 2023). While sporadic mutations may appear in zoonotic cases, established human circulation is characterised by a cumulative accumulation of specific TC→TT or GA→AA transitions driven by APOBEC3F (Suspène et al. 2023), a host’s restriction. To determine if the Sierra Leone epidemic fits this profile, we quantified mutational biases across the G.1 phylogeny by reconstructing ancestral single-nucleotide polymorphisms (SNPs). Our analysis reveals a substantial enrichment of these signatures, with approximately 85% (90/106) of reconstructed SNPs consistent with APOBEC3 editing activities.

Further, I think the manuscript would benefit from a formal assessment of APOBEC enrichment along lineages to determine when and where these signatures provide evidence of sustained human transmission. It is currently too descriptive.

We appreciate the reviewer’s suggestion regarding a formal assessment of APOBEC enrichment. Such analyses have been explored in detail in prior work (e.g., O’Toole et al., 2023; Parker et al., 2025); therefore, this study focuses on applying those established interpretations to the genomic and epidemiological context of the Sierra Leone outbreak.

We have revised the text to clarify the role of APOBEC signatures as supporting rather than primary evidence in this analysis.

Revised lines 201-204 now read, with a revised Supplementary figure 7:

“This dominant mutational bias provides strong genomic evidence that Lineage G.1 is the product of sustained human transmission rather than repeated zoonotic introductions observed in Clade IIa and zoonotic Clade IIb lineages (Figure 1C, Supplementary Figure 7).”

Supplementary figure 7: Comparative mutational spectra across MPXV lineages. Yellow bars indicate APOBEC3-signature mutations (C→T and G→A in TC/GA context), while grey bars represent all other substitutions. The massive enrichment in Lineage A.x and G.1 contrasts with the background mutation patterns seen in zoonotic lineages.

What kind of robustness checks can be applied to the inference that G.1 first became established in the Western Area Urban region. This region is the most represented in the phylogeny, and it is not clear to me that the analysis presented adequately addresses the authors' stated concern about "under-ascertainment of transmission elsewhere during the early phase of the outbreak".

We appreciate the reviewer's careful consideration of potential sampling bias, particularly the overrepresentation of the Western Area Urban (WAU) area. To assess whether our inference of establishment in WAU was robust and did not reflect a sampling bias, we evaluated this result using multiple complementary approaches:

- **GLM-based Assessment of Sampling Bias:** To explicitly test whether the identification of WAU as the epicentre was an artefact of surveillance intensity, we employed a Generalised Linear Model (GLM) within the phylogeographic framework. We included predictors specifically designed to capture sampling discrepancies, including location-specific sample sizes and binary origin predictors. Notably, our analysis rejected the hypothesis that sampling intensity drove the spatial reconstruction. The inclusion of location-specific residual effects was not supported (Bayes Factor [BF] ≤ 0.40). Furthermore, explicit binary predictors for WAU (the most sampled location) and Kenema (the second most sampled) were decisively rejected (BF ≤ 0.22). Instead, population density emerged as the strongest predictor of viral diffusion. This indicates that the high rate of viral export from WAU is driven by its demographic characteristics (high density facilitating transmission) rather than its overrepresentation in the sequencing data.

Generalised linear model coefficients for spatial spread covariates, with corresponding Bayes factors. Significant covariates ($BF > 3$) are highlighted in grey/red.

- Hierarchical Analysis (Regional Level):** As detailed in the results and method, we repeated the discrete phylogeographic reconstruction at the *regional* level. By aggregating sparse district data into broader administrative regions, we reduced the impact of heterogeneity at the district level; however, the Western region remains overrepresented, as shown in the figure. This analysis identified the Western Region (containing WAU) as the most likely source (Posterior Probability = 0.99; Supplementary Figure 3), consistent with the district-level results.
- Epidemiological Concordance:** We compared our phylogenetic inferences with independent epidemiological data. WAU reported the highest absolute case burden early in the outbreak (Figure 1B). The phylogenetic placement of the root in WAU aligns with the observed incidence curves and the known mobility network of the capital.

Comparison of cases and sequences across districts and regions sampled.

We have clarified in the Discussion that this represents a fundamental challenge in genomic epidemiology, particularly in settings where testing capacity varies significantly across districts. While we acknowledge sampling bias as a limitation, the convergence of results from our discrete district analysis, regional aggregate analysis, and GLM sensitivity analysis provides consistent support for Western Area Urban (WAU) as the primary hub of establishment. We have explicitly emphasised the necessary caveats associated with this inference in the revised text (lines 535-545).

“This study has several limitations, primarily related to the sample’s coverage and spatial heterogeneity. Although our dataset includes ~6.9% of confirmed cases, coverage varied widely across districts, ranging from fewer than 5% of cases in the high-burden Western Area Urban (131 of 2,867) and Western Area Rural (15 of 1,020) districts to more than 30% in Kenema, Bo, and Bonthe. Such uneven representation risks biasing phylogeographic inferences, potentially overestimating persistence or exports from better-sampled districts. To assess the robustness of our inferences against this heterogeneity, we performed a sensitivity analysis by aggregating districts into broader

administrative regions and testing for sampling artefacts using a generalised linear model. While these analyses consistently supported the Western Region as the origin (PP = 0.99) and rejected sampling intensity as a driver of diffusion, important caveats remain. First, due to lower sampling coverage outside the capital, our estimates of local persistence likely represent conservative lower bounds. Thus, while the shorter transmission chains observed in these districts may reflect true extinction events (sink dynamics), they may also mask undersampled local clusters. Our time-varying reproductive number estimates are further constrained by under-ascertainment, as suggested by the JUNIPER-estimated epidemic size of ~10,400 cases compared with the 5,096 confirmed by qPCR. Consequently, while the Western Area Urban district served as the primary node for amplification and dissemination, we cannot definitively exclude the possibility of cryptic establishment in an unsampled district or neighbouring country prior to its explosive detection in the capital.

I have similar concerns for the persistence analysis. How robust are these conclusions to biases in diagnosis rates and subsequent genome sequencing rates?

We acknowledge that heterogeneity in sampling intensity can influence the reconstruction of transmission chains. Specifically, lower diagnosis or sequencing rates in a given region increase the likelihood of "missing links" in the phylogeny, which causes what would be a single long transmission chain to appear as multiple independent, shorter introductions.

However, we believe our conclusions regarding persistence, and particularly the decline of persistence over time, are robust to these biases for two key reasons:

1. The Temporal Trend Contradicts Sampling Bias: Our analysis shows a strong negative correlation between epidemic time and transmission chain duration (i.e., chains became shorter later in the outbreak, Figure 4B). Notably, surveillance and sequencing efforts intensified as the outbreak progressed, particularly following the launch of the Enhanced Integrated Mpox Response (Figure 1A). If our results were driven by sampling artefacts, increased sampling intensity later in the outbreak should have resulted in *more* connected (i.e., longer) transmission

chains. The fact that we observe shorter chains despite improved surveillance strongly suggests this fragmentation is a true biological signal driven by public health interventions (isolation and vaccination) rather than a sampling artefact. Revised lines 420-435 now read:

“These transmission chains were predominantly seeded from the Western Area Urban district, with repeated introductions establishing multiple co-circulating chains with comparatively shorter periods of persistence in districts such as Kenema, Western Area Rural, and Bonthe (Figure 4A, C). The strong negative correlation observed between the duration of transmission chains and the time of the epidemic (Figure 4B) indicates that these transmission chains became shorter as the outbreak progressed. This pattern coincides with the timelines implementation of intensified public health interventions, particularly the vaccination campaigns that began in the Western Area Urban and Rural districts on March 26 and were expanded nationwide by April 30, alongside the ring vaccination of contacts initiated in early June (Figure 1A), helped reduce the susceptible population, the integrated phase of this intervention was initiated when it was observed that home-based management was unintentionally sustaining the transmission of the virus. The subsequent transition to mandatory treatment by isolation likely played a crucial role in breaking up transmission networks and preventing community clusters (Figure 4B).”

2. Persistence Estimates are Conservative Lower Bounds: In regions with lower sampling coverage (e.g., outside Western Area Urban), the bias described by the reviewer would lead to an underestimation of local persistence. Consequently, the transmission chains we identify represent a conservative lower bound of local circulation. The fact that we detect sustained persistence in Western Area Urban (the most heavily sampled district) provides evidence of established transmission there; conversely, the shorter chains in other districts may indeed reflect true repeated introductions (sinks) or undersampled local clusters. Importantly, our interpretation emphasises the dominance of Western Area Urban as a persistent source, a finding supported by both phylogenetic connectivity and independent epidemiological case data, as well as the additional GLM analysis.

Finally, to account for phylogenetic uncertainty, our Markov jump analyses were integrated over the full posterior distribution of trees, ensuring that our estimates of chain duration reflect the range of plausible evolutionary histories compatible with the data. Revised lines 342-348 in the result section now read,

“To investigate this pattern of within-country spatiotemporal spread while accounting for uncertainty in phylogeographic reconstruction due to sparse sampling, we incorporated a Markov jump counting approach. This method reconstructs the history of location changes along the phylogenetic branches, allowing us to estimate the timing and origin of geographic transmission chains (Lemey et al. 2009; Lemey et al. 2014), defined here as continuous periods of viral lineage circulation within a specific location following an introduction event both at the regional and district levels.”

The Results section lacks any indication of vaccine success/impact, the description of which the Discussion post-dates the genome sampling. I am left unclear on how to evaluate any statements regarding the impact of vaccines. Though I am wont to agree with the sentiment expressed in the Discussion, this paper does not provide evidence in favor of these conclusions.

We appreciate the reviewer’s important critique regarding the interpretation of intervention effects. We agree that our study design does not allow for a direct estimation of the causal efficacy of individual public health measures, such as vaccination. Consequently, we have revised the text to ensure our language reflects this limitation.

While our phylodynamic analyses provide a quantitative description of changes in transmission dynamics over time, we have reframed these findings to emphasise association rather than causality. Specifically, our analysis of transmission chain persistence (Figure 4B) indicates that transmission chains became shorter and less persistent as the outbreak progressed. In the revised Results section, we present this pattern alongside the timeline of public health interventions (Figure 1A), carefully noting that the observed fragmentation of transmission networks temporally coincides with the rollout of vaccination campaigns (initiated March 26 and expanded nationwide by April 30) and the implementation of mandatory isolation.

We explicitly state that these results describe changes in transmission network structure and are consistent with, but do not directly prove, reduced community transmission

resulting from specific interventions. We have removed language implying direct efficacy to align with the reviewer's correct observation.

Revised lines 424-435 now read:

“The strong negative correlation observed between the duration of transmission chains and the epidemic phase (Figure 4B) indicates that these transmission chains became shorter as the outbreak progressed. This pattern is consistent with the timeline of public health interventions. This notably includes the vaccination campaigns that began in the Western Area Urban and Rural districts on March 26 and expanded nationwide by April 30, alongside the ring vaccination of contacts initiated in early June (Figure 1A). While these campaigns reduced the susceptible population, an integrated phase was initiated after it was observed that home-based management was unintentionally sustaining viral transmission. The subsequent transition to mandatory isolation may have played a crucial role in breaking transmission networks and preventing community clusters (Figure 4B).”

Minor comments:

Figure 1A. I don't see an annotation about the start and/or location of vaccination campaigns. And the second y-axis is not labelled.

We thank the reviewer for their careful attention to detail. We have updated Figure 1A to explicitly annotate the timeline of key intervention strategies, including the start dates and locations of the vaccination campaigns (Western Area Urban/Rural and Nationwide). Additionally, we have added the missing label to the secondary y-axis to clearly indicate the time-varying reproductive number (R_t).

Revised Figure 1: A) Confirmed mpox incidence in Sierra Leone during 2025, showing the sequencing rate of this study over time (bars) and the time-varying reproductive number (line, right y-axis, with 95% confidence intervals). Annotations mark the start of vaccination campaigns in Western Area Urban (W.A.U.) and Western Area Rural (W.A.R.), as well as the launch of the national vaccination campaign

The description of Supplementary Figure 1 references a basal node corresponding with a Rotterdam sequence and a clade lying downstream of the 198 sequence. These descriptions, focused on sampled viruses rather than phylogenetic relationships are imprecise and inaccurate. I cannot reconcile this description with the tree depicted.

We thank the reviewer for this correction. We agree that our previous description relied on imprecise terminology that conflated tip dates with phylogenetic topology. We have revised the text to accurately describe the relationships depicted in the tree: specifically, that the Sierra Leone–Guinea cluster forms a sister clade to the lineage containing the 1965 Rotterdam isolate, and that this broader group shares a common ancestor with the clade containing the 1958 Copenhagen sequence, rather than being "downstream" of it. Supplementary Figure 1 (lines 941-962).

“Supplementary Figure 1: Clade IIa phylogeny with reconstructed SNPs mapped onto branches. The tree shows the phylogenetic placement of the novel Sierra Leone Clade IIa sequence (red tip, enlarged) within a cluster of recent Guinea genomes. This cluster is a sister to the historical Rotterdam/Orangutan lineage. Ancestral state reconstruction was performed to map single-nucleotide polymorphisms (SNPs) to branches. APOBEC3-characteristic substitutions (C→T or G→A in the TC/GA context) are colored yellow, while all other substitutions are colored grey. The tree is rooted to the novel Clade IIb zoonotic outgroup identified in Parker et al. (2025).

”

Response to Reviewers for Campbell et al.

Reviewer #1 (Remarks to the Author):

The author appropriately addressed all comments. I have a problem with the following statement:

"However, there is a difference among these Clades in terms of clinical severity. Clade I, for example, has historically been associated with higher case fatality ratios (CFRs) than clade II (Cho et al. 2024; Beer and Rao 2019),(Beer and Rao 2019; Wawina-Bokalanga et al. 2025). Recent trial data from the DRC highlight differential patient outcomes with the recommended tecovirimat drugs, unless patients are hospitalised and receive high quality supportive care. (Postal et al. 2025; PALM007 Writing Group et al. 2025) The clinical severity of Clade IIa infection in humans is still less well characterised than that of Clades Ia/Ib and IIb.""

Although Clade I has historically been associated with more severe disease, more recent data indicate important heterogeneity within this lineage. Accumulating epidemiological evidence does not demonstrate remarkable mortality differences between Clade Ib and Clade IIb. In the PALM tecovirimat trial, Clade Ia mortality was 1.7%, markedly lower than the ~10% CFR frequently cited in earlier literature. Reported mortality for Clade Ib has generally remained below 1% in most affected countries.

While mortality may be higher for Clade Ia compared with other subclades, there remains considerable uncertainty as to whether this reflects intrinsic viral virulence or differences in the affected populations. Clade Ia transmission has predominantly occurred among children in endemic regions, where contextual factors—such as malnutrition, co-infections, delayed healthcare access, and limited supportive care—may substantially influence observed outcomes.

Experimental laboratory data do suggest greater virulence of Clade I viruses relative to Clade II. However, contemporary epidemiological data do not consistently demonstrate large differences in clinical severity across subclades, particularly when healthcare access and supportive management are accounted for.

For accuracy and precision, it would therefore be preferable to modify the sentence to explicitly refer to Clade Ia rather than Clade I as a whole, reflecting the comparatively low reported mortality associated with Clade Ib.

We thank the reviewers for this observation. We agree with the reviewers and have updated it to reflect this nuance.

(Line 124-128)-“ However, clades differ in clinical severity: Clade Ia has historically been associated with higher case fatality ratios (CFRs) than Clade IIb, though recent data suggest this partly reflects contextual factors, such as malnutrition, co-infections, and limited healthcare access, rather than intrinsic viral virulence alone, and reported mortality for Clade Ib has remained below 1% in most affected countries, comparable to Clade IIb.^{12,25,25,26}

(Line 129-130)-Recent DRC trial data further highlight that tecovirimat outcomes vary substantially unless patients receive high-quality supportive care.^{27,28} The clinical severity of Clade IIa in humans remains less well characterized than that of the other subclades.”